# Living and Coping with Olfactory and Taste Disorders: A Qualitative Study of People with Long-COVID-19

**DOI:** 10.3390/healthcare12070754

**Published:** 2024-03-30

**Authors:** Paloma Moro-López-Menchero, María Belén Martín-Sanz, César Fernandez-de-las-Peñas, Stella Maris Gómez-Sanchez, Antonio Gil-Crujera, Laura Ceballos-García, Nuria I. Escribano-Mediavilla, Mª Victoria Fuentes-Fuentes, Domingo Palacios-Ceña

**Affiliations:** 1Department of Physical Therapy, Occupational Therapy, Physical Medicine and Rehabilitation, Research Group of Humanities and Qualitative Research in Health Science, King Juan Carlos University, 28922 Alcorcón, Spain; paloma.moro@urjc.es (P.M.-L.-M.); domingo.palacios@urjc.es (D.P.-C.); 2Department of Physical Therapy, Occupational Therapy, Physical Medicine and Rehabilitation, Research Group of Manual Therapy, Dry Needling and Therapeutic Exercise, King Juan Carlos University, 28922 Alcorcón, Spain; cesar.fernandez@urjc.es; 3Research Group GAMDES, Department of Basic Health Sciences, King Juan Carlos University, 28922 Alcorcón, Spain; stella.gomez@urjc.es (S.M.G.-S.); antonio.gil@urjc.es (A.G.-C.); 4Department of Nursing and Dentistry, IDIBO Research Group, King Juan Carlos University, 28922 Alcorcón, Spain; laura.ceballos@urjc.es (L.C.-G.); nuria.escribano@urjc.es (N.I.E.-M.); victoria.fuentes@urjc.es (M.V.F.-F.)

**Keywords:** olfactory disorders, taste disorders, anosmia, COVID-19, post-acute COVID-19 syndrome, patients, qualitative research

## Abstract

Taste and smell disorders are common symptoms of SARS-CoV-2 acute infection. In post-COVID-19 condition, symptoms can persist leading to disruption in patients’ lives, to changes in their coping skills, and to the need to develop strategies for everyday life. This study aimed to describe the perspective of a group of patients with Long-COVID-19, a condition where loss of taste and/or smell was the most predominant symptom. A qualitative descriptive study was conducted. Participants who had suffered SARS-CoV-2 infection and had Long-COVID-19 loss of taste and/or smell were recruited. Purposive sampling was applied, and participants were recruited until data redundancy was reached. In-depth interviews were used for data collection and thematic analysis was applied. Twelve COVID-19 survivors (75% women) were recruited. The mean age of the participants was 55 years, and the mean duration of post-COVID-19 symptoms was 25 months. Three themes were identified: (a) Living with taste and smell disorders, describing the disorders they experience on a daily basis, how their life has changed and the accompanying emotions, (b) Changes and challenges resulting from the loss of taste and smell, changes in habits, self-care and risk in certain jobs or daily activities, (c) Coping with taste and smell disorders, describing the daily strategies used and the health care received. In conclusion, Long-COVID-19 taste and/or smell disorders limit daily life and involve changes in habits, meal preparation, and the ability to detect potentially dangerous situations.

## 1. Introduction

People suffering from coronavirus disease, 2019 (COVID-19) experience symptoms including dry cough, dyspnea, fatigue, fever, diarrhea, anorexia, nausea, vomiting, headaches, and/or muscle pain [1,2,3]. However, a distinctive feature of severe acute respiratory syndrome coronavirus 2 (SARS-CoV-2) acute infection is the presence of smell and taste disorders [1,4,5]. These olfactory disorders may be present after the acute infection, as a part of what is known as post-COVID-19 condition [6].

Post-COVID-19 olfactory and taste disorders include partial loss of smell (hyposmia), loss of taste (hypogeusia), total loss of smell (anosmia), or total loss of taste (ageusia) [5]. Additionally, they may be accompanied by dysgeusia (persistent bad taste in the mouth, rancid or metallic) or chemesthesis (the ability to detect chemical irritants), parosmia (distorted sense of smell in the presence of an odor source), and phantosmia (distorted odor perception in presence of an odor source) [7,8]. Previous systematic reviews and meta-analysis studies [9,10,11,12,13,14] showed different prevalences of taste and smell dysfunction. Thus, taste dysfunction in patients with COVID-19 affected between 15% and 81.6%, while olfactory/smell dysfunction presented intervals of 10–74.8% [9,10,11,12,13,14]. Finally, studies that examined the presence of both symptoms reported intervals of 18.6–35.04% [9,12]. Loss of taste after viral infection was unusual, and it was hypothesized that the loss was caused by loss of smell [2,15]. However, it has been shown that the loss of taste represents a separate symptom [15].

To explain the pathophysiology of olfactory and taste disorders due to SARS-CoV-2, three mechanisms have been proposed: (1) conduction loss caused by olfactory cleft obstruction; (2) lesion of the olfactory epithelium; (3) lesion of the olfactory bulb [16]. At present, research is continuing and there has been no fully defined explanation since SARS-CoV-2 does not cause nasal congestion or secretion [2,17,18]. Previous research [4,17] suggests that loss of taste and smell is caused by a high viral load in the nasal cavity resulting in a loss of function of olfactory sensory neurons and taste buds, caused by infection, inflammation, and subsequent mucus dysfunction.

There is no clear prognosis or treatment for smell and taste disorders [2]. However, nasal saline washes, oral or spray corticosteroids, vitamins, and olfactory rehabilitation exercises are the most used tools for recovery [5,19]. Philpott et al. [20] conducted a study on the experience and self-care of patients with olfactory disorders, reporting that of those who received treatment (*n* = 42), in 96% treatment was ineffective, and of those who did not receive treatment (*n* = 58), a decrease in their perceived quality of life and an increase in mental health problems were perceived in almost all cases.

Previous observational studies demonstrated that people who suffer from post-COVID-19 taste and smell disorders suffer a decline in their daily and social quality of life, affecting their eating habits and customs, and negatively impacting their physical and mental wellbeing [9,21]. Furthermore, there are few qualitative studies [22,23,24,25] on the experience of patients describing their loss and/or alterations of taste and smell due to SARS-CoV-2 acute infection. The available studies [22,23,24,25] report how this condition has decreased their ability to detect hazards such as toxic odors, smoke/fire, or gas leaks. Furthermore, distortions of smell have a direct impact on the experience of eating and enjoyment of food since the taste of food is the result of the combination of smell and taste [22,23,25].

The presence of symptoms that are perpetuated over time and can become chronic conditions affects the perspective of patients and their families. Also, it influences their expectations of cure, adherence to treatment, and the development of strategies to enable them to adapt to the symptoms and changes caused by the disease [26]. There is limited literature available to date describing patient perspective and experience of Long-COVID-19 loss of taste and smell [8,27]. Almgren et al. [27] (Sweden) and Burges Watson et al. [8] (UK) reported how olfactory and taste dysfunctions in patients with Long-COVID-19 decreased self-confidence, leading to a decreased desire to eat and impaired their ability to prepare food and caused changes in body weight, which may even affect patients’ self-perception and self-esteem, leading to an altered relationship to themselves and others.

However, further studies are needed to describe the experience of living with long-lasting loss of taste and/or smell among people with Long-COVID-19 as well as the consequences of these disorders and how patients cope and develop strategies to deal with them in different geographical and cultural contexts. The guiding question for this study, based on the EPPIC framework, was as follows [28,29,30]: What is the perspective of people with Long-COVID-19 with taste and smell disorders who belong to Spanish Long-COVID-19 patient associations? In addition, how was the daily life of these people? What challenges and obstacles did they face? How did they deal with these difficulties? The aim of this study was to describe the perspective of a group of individuals with Long-COVID-19 taste and smell disorders regarding their daily life.

## 2. Methods

### 2.1. Study Design

A descriptive qualitative study, based on an interpretative paradigm [28,31,32,33] according to the Standards for Reporting Qualitative Research [34] and the Consolidated Criteria for Reporting Qualitative Research [35] was conducted. Descriptive qualitative designs are based on the participants’ own words to describe their personal experience, as well as their perspectives on certain phenomena such as illness and disability [31,36,37]. The aim of a descriptive qualitative study is to identify an event (disease, health problem, symptoms, condition) or a critical situation [28,31,32]. This type of qualitative design enables the use of a wide range of data collection instruments, and different proposals for analysis, to describe what is happening and how is it happening from the perspective of those involved in their own environment [32,33]. Qualitative descriptive studies aim to be a comprehensive summary of events in the everyday terms of the described event [33,36,37]. This design is the method of choice when straight descriptions of phenomena are desired [33,36,37]. The current study was approved by the ethics and research committee of the University Rey Juan Carlos (Code: 0907202015920). Written consent was requested prior to inclusion of participants and data collection. The study also adhered to the Helsinki Declaration.

### 2.2. Participants, Context, and Sampling Strategies

Purposive sampling was applied to select participants with relevant information for the study [38]. This type of sampling strategy focuses on deliberately recruiting a specific group of people who have experienced a phenomenon of interest [38]. Thus, participants were recruited from two COVID-19 patient organizations (Long-COVID ACTS and COVID persistente España—Persistent COVID Spain) through flyers, social networks, and internet platforms for COVID-19 support groups. All participants from the two associations who voluntarily wished to participate were included consecutively. The inclusion criteria consisted of COVID-19 survivors, aged over 18 years with long-lasting post-COVID-19 olfactory and taste symptoms (total or partial loss)—confirmed by medical specialist, otorhinolaryngologist—for at least three months duration after COVID-19 diagnosis, as confirmed by the positive reverse transcription–polymerase chain reaction test from a nasopharyngeal or oropharyngeal swab.

Participant recruitment and data collection ceased when thematic saturation was reached, meaning that the information obtained from the interviews became repetitive (no new themes emerge from the data) [38,39,40]. The authors agreed that information would be considered repetitive when there was sufficient in-depth data showing themes, categories and patterns of the phenomenon studied. The iterative process began by conducting the first interview, transcribing and analyzing it to identify the themes (see Section 2.4. Data Analysis). Subsequently, the second interview was conducted, and the same process (data collection–transcription and analysis) was carried out until the themes were derived from the data. The themes of both interviews were then compared to identify whether there were new contents. If there was new content, it meant that it was necessary to continue collecting data on the phenomenon under study and the next interview was conducted. This process continued with each interview, progressively comparing the results of the analysis of each one, until no new themes were identified. To this end, the authors reviewed the analysis and the quality of the participants’ quotes [38]; in the current study, this situation occurred after including 12 participants.

### 2.3. Data Collection

Data collection occurred from May 2022 to October 2022. Data collection included semi-structured interviews, using an open-ended question guide focused on research areas, and field notes from the researchers (Table 1) [41].

During the interview, the researchers noted key words and topics and retrieved these using related questions to clarify the content [41]. Additionally, the expression “Please, tell me about that” was used (if necessary) to increase the depth of the discussions surrounding specific topics in each interview. Interviews were conducted via the Microsoft Teams digital platform [42], which supported audio and/or video recording. A total of 496 min of interviews was recorded. The shortest interview lasted 24 min, and the duration of the longest interview was 1 h and 33 min. The interviews were conducted by PMLM, MBMS, SMGS, and AGC, who completed the information (as a secondary data source) with their field notes.

### 2.4. Data Analysis

The in-depth interviews and the researcher’s field notes were transcribed verbatim. A thematic analysis of these texts was carried out by identifying the text fragments that could provide relevant information [43]. Based on the text fragments, the codes containing the most descriptive information were identified. Subsequently, the codes were grouped into categories according to the similarity of their content [43,44]. Subsequently, themes were identified from the categories, reflecting the experience of the participants. To identify the themes, the results were combined through joint team meetings, in which the final themes were presented, combined, integrated, and identified. In the event of disagreements, team discussions were held to reach a consensus. No qualitative software was applied for data analysis.

### 2.5. Rigor

The criteria for trustworthiness developed by Guba and Lincoln were applied [45,46]. The techniques used to monitor trustworthiness are listed in Table 2. The use of these methods to increase rigor is compatible with the descriptive qualitative design [31,32].

## 3. Results

Twelve COVID-19 survivors (75% women, age interval: 39–66) were recruited. All participants had taste and smell disorders after acute SARS-CoV-2 infection and only two participants were receiving some type of treatment during the study (18.8%). Patients reported post-COVID-19 smell and taste disorder symptoms from 25.25 months. See Table 3. The professions and jobs of the participants were as follows; (a) school teacher (*n* = 1), retired person (*n* = 1), nurse (*n* = 2), nursing assistant (*n* = 2), retail clerk (*n* = 1), cleaner (*n* = 2), engineer (*n* = 1), and office-administrator (*n* = 2).

### 3.1. Theme 1. Living with Taste and Smell Disorders

Three themes emerged from the data analyzed with their corresponding categories: (a) Living with taste and smell disorders; (b) paths and challenges resulting from loss and/or impairment of taste and smell; (c) coping with taste and smell disorders. See Table 4. Participant narratives were integrated into the results. The inclusion of narratives ensures the credibility of the results presented [46].

#### 3.1.1. Living with Absent and/or Impaired Sense of Smell

Our participants recounted how their quality of life was reduced by the loss of smell. Most mentioned that their sense of smell was not as good as before, that they would have to learn to live with it, but that it was not something that they were overly fixated with or that they considered to be a big problem. Some participants, however, said that this loss caused them to feel socially isolated: “… *the loss of senses leaves you a bit isolated [...] if you don’t smell or taste... well, socially and in terms of everything, you are isolated.*” (*P5*).

They reported that sometimes they detected strange odors that were difficult to identify. One of the participants reported that she was happy when she suddenly smelled something, since it happened very rarely, even if it was a strange odor or one that was difficult to identify. Others described how smells seemed fainter, thus they needed strong smells to be able to identify them. The participants reported that they were unaware of the smell, as if they forgot to smell: “… *It smells, and all of a sudden I stop smelling and I forget.*” (*P10*).

The participants expressed feelings of sadness and grief, anger, together with frustration at the uncertainty of not knowing if they were going to recover their sense of smell.

#### 3.1.2. Living with Absence and/or Altered Taste

The loss of the sense of taste was related to more tangible and concrete repercussions for the participants. They gave clear examples of how this loss could cause complications in the future, oriented to food consumption, preparation, and selection (avoidance of food consumption, difficulty in assessing the state of food when cooking and buying it).

The loss of taste was a source of great frustration for participants. They did not feel like eating because everything seemed to taste the same, they had trouble identifying what they were eating, and they even avoided food at times. The participants stated that it is key for them to feel the taste of what they eat and, above all, to perceive that there are differences between the foods and the different meals. For those individuals who partially recovered their sense of taste (P5,P7,P8,P9), they found it strange or unpleasant to consume certain foods and beverages because they did not taste anything or the taste was different from before, and therefore they ceased to enjoy it: “… *It’s a strange feeling, you drink a Coke, and it doesn’t taste of anything, you eat something, and it doesn’t taste of anything.*” (*P7*).

The participants stated that they could not taste, identify, or assimilate food well. They reported that certain ingredients, such as onion or bell pepper, distorted the taste of food. In addition, they found it strange not to be able to differentiate sweet, salty, bland, bitter, or sour. They also reported that they were able to taste foods they considered to have a stronger taste, such as watermelon or tangerines: “… *Some foods I don’t taste at all, but they are foods with less flavor. For instance, a tangerine, or an orange... these are more intense flavors that I do taste, not 100%, but I am able to taste them.*” (*P9*).

Nevertheless, most participants agreed that a metallic taste in the mouth prevailed over anything else.

#### 3.1.3. The Meaning of Loss of Sense of Smell and Taste

The participants described how the absence of smell went from not smelling anything, to having “olfactory” bursts, describing that the smell disappeared and appeared, thus, it kept coming and going. This total absence of smell, together with the sudden appearance of olfactory bursts disoriented the participants because they were not able to know whether they were sick or not or whether the symptoms meant a bad or good prognosis. This presence and absence was experienced with great uncertainty: “*I can’t say I don’t smell because I smell, not when I want, not what I want, but I smell, I don’t know when or where, but it hasn’t disappeared... I can’t tell you if I’m sick…*” (*P7*).

The absence of smell and taste was experienced by some participants as a loss, a disconnection, since through smell they had been linked to many aspects of their daily lives (meeting people for dinner) and smells had been associated with important moments in their lives, such as people or memories. They felt that they had lost a gateway or a door to those moments that are part of their life: “… *Smells and taste stimulate many things and mark a person’s life. Many memories of my life are associated with sensations, smells, and tastes. Like the food that my mother, who passed away, used to cook for me. It’s something I’ve lost, that disconnects you from your past.*” (*P9*).

Most of the participants claimed to live on the memory of smells and tastes. They acknowledged that they remembered or imagined the smell and taste of foods or products such as gasoline or cologne to identify them. For them, these symptoms have a strong relationship with their mind, their brain: “…*the smell is in the mind [...] the brain has a memory, and you relate it to the enjoyment of eating.*” (*P9*).

The participants felt resigned, and many considered that they would not be able to return to a normal life after losing their sense of smell and taste. Some narrated feeling mutilated, as if they had lost a limb or some part of their body. The meaning they give to the symptoms is the loss of something precious: “… *You become mutilated because they have removed a piece of... there is an area of your body that no longer functions [...] well, things that always remind you that you have a mutilation here in the nose or in the brain.*” (*P3*).

#### 3.1.4. Low-Priority Senses

It is paradoxical how the participants experienced these symptoms with a sense of loss, disconnection with their past, with their environment, and related these symptoms to other organs, such as the brain, while at the same time not giving them enough importance to be considered as symptoms that imply seeking immediate professional help.

The participants considered taste and smell to be the least important senses. Thus, they related that the lack of taste and smell was less important or disabling than other health problems or loss of other senses such as sight. Therefore, they neglected this loss because they considered that they could do without these senses and because, compared to everything that COVID-19 could cause, it was just a minor sequela of the virus: “… *It is not as disabling as, for example, tiredness, fatigue, or cough, which made me unable to speak.*” (*P11*).

### 3.2. Theme 2. Paths and Challenges due to the Loss and/or Alteration of Taste and Smell

#### 3.2.1. Changes due to Loss and/or Alteration of Sense of Smell

Although both senses were lost or affected, and the consequences of the loss of taste were closer and more tangible for our participants, the most affected sense was the sense of smell. This forced the participants to change their habits and was detrimental to their social or leisure relationships as they were more cautious and less inclined to socialize with their friends. They described how the loss of smell and taste isolated them socially: “*The main change I noticed was that I began to stop doing things, I changed my routines, there were things that didn’t really make sense for me to do, like drinking or eating with friends, I didn’t taste anything I ate... I preferred to be alone.*” (*P3*).

Moreover, the loss of sense of smell made them feel insecure and dependent. Participants reported that they lived in fear, due to the danger of not detecting a gas leak, a fire, cooking and having their food burned, or suffering food poisoning because of the inability to smell food: “… *you’re going to laugh, but not being able to smell, or not being able to trust that you smell something correctly, or that you may confuse the smell, makes me feel weak, insecure…*” (*P8*).

In addition, some people acknowledged that they were unable to detect whether something smelled good or bad, as their perception of scents had changed (or disappeared), and some things smelled different, or unpleasant. They also stated that they tended to confuse strong smells such as rotten fish and garbage: “*Often, I find that I confuse smells or exchange them, for example sometimes a plate of food with strong spices smells like garbage and vice versa. It’s strange that garbage smells like something I would like to eat, isn’t it?*” (*P2*).

The loss of smell also affected the participants’ body hygiene. They were forced to change their toiletries because their perception of their own body odor had changed. Deodorant smelled bad, they were worried about not smelling their sweat, not smelling their children’s perfume, or not knowing if they had put on too much perfume themselves. For this reason, many participants claimed to have stopped using perfumes and colognes because they no longer smelled the same: “… *I have worked a lot with perfumes, because it is one of my favorite things and hobbies, and I used to wear colognes, but now I forget to wear them, because I no longer enjoy them.*” (*P12*).

#### 3.2.2. Work-Related Difficulties of Olfactory Loss

Most participants did not have any problems at work, except for those who, due to their professions, could experience limitations or risks when performing their work with a loss of smell (P3, P7, P10). The distortion or absence of smell prevented certain participants from detecting specific situations such as the need to change the diaper of small children in a nursery school (P1), the diagnosis of some diseases with very characteristic odors (P3), and even the risk of intoxication by chemicals or cleaning products (P7): “… *My job is to install security systems and I have to smell the hydrogen sulfide, if I don’t recover my sense of smell, I can’t get back to work because of the risk of intoxication.*” (*P7*).

However, one participant was a deviant case (P6), stating that anosmia was positive when he had to sporadically perform his professional duties at the landfill.

In cases where the sense of smell was directly related to their work, no participant considered the possibility of abandoning their job and looking for a new one that did not depend on that sense. For them, it was not a real possibility worth considering because of the economic repercussions it would have for the participants and their families.

#### 3.2.3. Changes due to the Loss and/or Alteration of Taste

The participants’ eating habits also changed, as their taste was not as sharp as before, although they were able to distinguish some stronger flavors. Some foods triggered more rejection than before, such as ice cream, fruit, vegetables, or candy, because they stated that they tasted worse.

Likewise, changes in eating habits were also caused by not feeling like eating because participants were frustrated about not tasting or enjoying the food and the discomfort caused by the limitation of not fully enjoying what a normal meal offers: “… *I don’t experience food the way I used to. Eating was not only eating, it was also feeling through food. Now it’s like eating cardboard, it’s no good.*” (*P7*).

A consequence of the alteration and/or loss of taste is the feeling of insecurity when assessing food. Many participants acknowledged feeling less confident in their ability to perceive whether food was in good condition when buying, preparing, or eating it.

There were deviant cases in our participants, where some described positive consequences of taste disorders (P4, P12) such as quitting drinking alcohol, or eating new foods that they did not consume before: “…*Now I can eat foods that I couldn’t stand before because they don’t taste the same to me or one thing tastes the same as the other, so I take advantage of this and eat more fish, for example, which I don’t like.*” (*P12*).

### 3.3. Theme 3. Coping with Taste and Smell Disorders

#### 3.3.1. Daily Strategies

In general, participants placed products and/or foods very close to their noses, to try to detect the smell and/or taste. As a result, they adopted positions and made gestures that appeared strange to those around them. Over time, these gestures eventually became integrated into the habitual body dynamics of the individuals in order to be able to smell, something which they were not always aware of. This led to “odd” situations with them sniffing everything: “… *I get very close to a perfume and take a deep breath to smell it. Sometimes they look at me like, “What are you doing? Like I’m a freak.*” (*P9*).

Regarding personal hygiene, the frequency of bathing/showering was increased, and the routine use of cologne was ensured to avoid body odor. When in doubt as to whether they smelt good or bad, the participants, being unsure, always opted to bathe again. Another strategy employed when using colognes, is that they ask people they trust in their social circle (friends, partners, family) to smell them, to make sure they do not smell unpleasant: “… *My wife is very attentive to me, making sure that I wear perfume to go out or that I don’t wear it twice.*” (*P4*).

Some participants told us that sometimes they achieved the opposite outcome when using cologne or perfume. As with bathing, when in doubt, they tended to use too much perfume or cologne, resulting in a more potent, overpowering scent, which had the opposite effect.

Regarding food, the participants stated that they ate by sight, since they needed to look at the food to identify its taste. This meant that when they had to eat foods they did not know or identify, they preferred not to eat. Another strategy used on these occasions, is that they depend on the recommendations of third parties: “…*I made ceviche, and my daughter said: “it’s delicious, it came out really good, it was tasty”, and so I felt the same way.*” (*P4*).

Another strategy they acquired was to actively participate in shopping and preparing food, to be able to appraise the product and buy it on the same day, and become more involved in the kitchen and in the preparation of food. When preparing meals, they varied the seasonings and ingredients to be able to identify the flavors, and even resorted to strong flavors to appreciate and enjoy food: “*If I wanted to feel confident about what I was eating, I had to cook it. So, a lot of doubts would disappear, and I would prepare it in a way that I knew I could eat it and that I was going to taste the food. … For meals there must be strong flavors, which is what I identify.*” (*P6*).

Regarding the use of salt, they opted for two strategies. They either stopped using salt in their meals to avoid the risk of overdoing it, or they invented ways of measuring the amount, such as using a spoon or containers, for each recipe. Nevertheless, they always sought help from people nearby to verify whether they had added the right amount of salt, or whether food was good to eat.

At work, participants described how they relied on coworkers to recognize odors, and were guided by the perception of their peers. P3 said to her partner: “*The truth is that I can’t smell a thing, thank goodness you’re here...*”

One participant reported avoiding specific activities for fear of getting intoxicated with chemical products.

#### 3.3.2. Health Care

Only a few participants (P3, P4, P5) described having received treatment or updated information about their problem. This forced them to do their own research or to follow the recommendations of third parties. The patients narrated that they were unable to determine which knowledge obtained from the Internet was the most appropriate or safest based on scientific knowledge.

On some occasions, patients did not receive specific treatment because their public referral hospitals did not have dedicated units for the management of Long-COVID-19. This required patients to seek help from the private health sector. Some participants described not being able to undergo treatments because these were private (not covered by public health care) and they could not afford the expense or because they could not afford to miss work to attend rehabilitation: “… *Therapy hours are incompatible with work hours. I have to choose between quitting my job and quitting therapy.*” (*P4*).

There were several ways to access treatment: (a) from the public sector through the primary care physician; (b) from the private sector through specialists such as the otorhinolaryngology specialist; or (c) derived from other work specialists, such as the occupational health physician.

Among the treatments received for the sense of smell, our participants highlighted the use of inhaled and oral corticosteroids, vitamins, and rehabilitation. Rehabilitation consisted of smelling different products, spices (oregano, cinnamon, or thyme, etc.), flowers, and trying to differentiate between them with the eyes closed, and in different concentrations. Many participants had received multiple treatments and diverse therapies (evidence-based and non-evidence-based), resulting in the feeling that no one and nothing worked for their problem: “… *the therapist told me: <you have to be in a state of calm because the brain has to be calm to recognize, to recover the sense of smell>. Perfect, but when I asked him if he could explain in more depth the concept of the brain calming down and how that activates my olfactory pathways, I was not confident with the response…*” (*P5*).

As a result, the treatment was perceived as disappointing because of the limited changes incurred in their daily lives.

Another key aspect was the role of the healthcare professional as a source of support and guidance for the patient. On some occasions, the fact that health professionals themselves trivialized anosmia, downplaying its importance because it was not a “disabling” condition, was perceived as a barrier for some participants when pursuing any type of treatment: “… *That’s the only barrier I find, that [physicians] trivialize the subject. It’s tough to be very worried, to share that worry and then to find that the person who is supposed to help you doesn’t pay attention to it.*” (*P3*).

The patients pointed out comments that they felt professionals should not use, such as the following: (a) questioning the degree of loss of smell or taste “but are you sure you can’t smell? have you tried?” (b) comparing their condition with other COVID-19 patients “you could be worse”, “it’s not that bad”, “there are people who are worse after COVID”; (c) giving information that discourages their efforts to resort to other treatments for recovery, “the best thing is that you do nothing, medicine only goes so far, accept it, it won’t do you any good”; (d) judging their decisions, “did you really spend your money to do that? don’t you know that it’s not good for anything? You’re making a mistake”.

## 4. Discussion

Olfactory loss or disorder is characterized by a fluctuating course. Burges Watson et al. [7,8] in their study on the impairment of taste and smell by Long-COVID-19, described how there are fluctuating changes in taste, and unstable recoveries, which cause uncertainty in patients based on the evolution of their symptoms. In addition, these authors [8] described the lack of smell and taste as an “invisible” condition where its consequences are not easily identifiable, nor recognized by patients or professionals. The loss and/or alteration of taste and smell can lead to nutritional problems, derived from lack/excess of food, change of habits (food intake and frequency), difficulty in food acquisition and preparation, etc. [8]. We believe that the fluctuating symptoms and their low intensity could affect the early detection of the disease, its follow-up, and the monitoring of its effects.

According to our results, the loss and/or disorders of smell and taste were not considered a priority for COVID-19 survivors. It is paradoxical how the absence and/or alteration of these senses is experienced as a loss but is not accompanied by a behavior of seeking professional help, or even resigning oneself to living with the lack of these senses. A possible explanation appears in the work of Almgren et al. [27], where after the loss of smell and taste, Long-COVID-19 patients compared themselves with other patients who had suffered COVID-19, compared symptoms and sequelae, and concluded that their problem was not so serious, and they should feel grateful for continuing to maintain good health free from severe disabilities.

Our results show how the absence and/or alteration of smell and taste can lead to situations of social isolation. This could be an attempt to minimize the risk of having episodes of nausea and vomiting in front of people, and the lack of pleasure with eating [23]. The authors believe that social withdrawal is an attempt to withdraw from situations that they may consider socially awkward such as confirming whether they liked a meal/dinner, or not being able to recognize smells and tastes when tasting a food or beverage (wine tasting). In addition, the presence of sadness and frustration may perpetuate and stimulate this social withdrawal. Previous studies [1,5,49] concur that people with post-COVID-19 taste and smell alterations experience a negative effect on social, occupational, and mental well-being dimensions. In addition, Alshakhs et al. [21] described cases of social isolation and disinterest regarding activities of daily living, as reactions of patients to the persistence of problems with taste and smell. The disorders of taste and smell caused by COVID-19 are linked to negative emotions and mood swings [8,50]. In contrast, Almgren et al. [27] in their study on symptoms in patients with Long-COVID-19, described how although patients felt guilty when they refused an invitation or social encounter, they progressively lost the guilty conscience when turning them down and became progressively more confident in saying no.

In addition, previous studies reported difficulties for people to track their own personal hygiene and body odor [8,19,49,51]. Burges Watson et al. [8] described the reasons why the loss and/or alteration of smell have important repercussions in the intimate and social dimension of patients with Long-COVID-19. These authors [8] described the importance of smell (and odor) with desire, and intimate (sexual) relationships. The symptoms caused difficulties in maintaining satisfactory sexual and intimate relationships with other people. Not only romantic and sexual relationships were affected, but also maternal bonding with babies and children. In our results, no participant mentioned an effect on their intimacy and sexual relations or direct effects on their relationship with their partner. Among the strategies to avoid bad odor, patients with Long-COVID-19 showered frequently, changed their clothes multiple times, or washed continuously [8]. In addition, asking for reassurance when checking their body odor, or consulting whether they are wearing enough perfume or aftershave, were other commonly used strategies [19,49]. We believe that the application of certain strategies, such as odor control through products or asking others for help in confirming their odor, allows patients to regain some sense of control over their symptoms [8,27].

The consequences of the loss of taste and smell include changes in habits and customs. Among these changes, participants avoided food or ate unwillingly. One of the reasons for this behavior is that they do not fully enjoy the eating experience, not savoring the food. In this line, Arndal et al. [50], Turner and Rogers [22], and Elkholi et al. [49] highlighted how after suffering COVID-19, patients showed a loss of appetite and a disinterest in food, due to a decreased enjoyment of food, reduced ability to savor food, and less pleasure dervied from eating. Decreased intake, however, does not always occur. The loss and/or alteration of taste and smell can cause unexpected effects such as increased intake or no change in appetite [22]. In addition, the beneficial or detrimental effect on health would depend on the increase or decrease in the consumption of certain foods. Thus, Turner and Rogers [22] noted that in some post-COVID-19 patients, there was an increase in the consumption of alcoholic beverages, due to the fact that, as the flavor was less intense, it was easier to over drink. In addition, Burges Watson et al. [8] described how to enhance taste sensation that they frequently consumed potato chips, chocolate, chili chips, and other foods that provided unusual textural experiences. Previous studies showed that patients who suffered COVID-19 present difficulties preparing food, eating foods that are more flavorsome, adding salt, pepper, sugar, or spice to meals, and using different textures [8,22,49]. In contrast, we believe that symptoms do not always cause negative effects on the participants’ health (alcohol consumption or reduction of vegetables), since our results show that there were cases in which the consumption of fish improved or the consumption of sweets decreased. An individualized nutritional assessment would be necessary to evaluate the effect of the dietary changes derived from the symptoms. In addition, using health education, patients could be taught to adopt food quality control strategies, and their involvement in the purchase and preparation of their own meals could be advocated. In this manner, their own control would be encouraged in the face of the feeling of uncertainty associated with certain symptoms.

Previous studies pointed to the exposure of patients with post-COVID-19 olfactory loss involved in potentially dangerous work situations, for instance, perfumers, firefighters, or chefs, who require their sense of smell to perform their duties [19,50]. Conversely, Elkholi et al. [49] showed that the loss of smell could become an adaptation strategy in certain jobs (landfills), as they do not have to protect themselves from strong or unpleasant odors. Our findings show that participants with jobs linked to smell have difficulties, but in no case do they consider quitting their jobs. We believe that those with jobs where smell and/or taste are required, and who suffer Long-COVID-19 symptoms, could avoid or delay informing their superiors and/or occupational medicine service of their situation, due to the economic repercussions that could result from their relocation or (temporary) termination from their job. In addition, the adaptation of work to the sequelae or symptoms of loss of smell and/or taste in those affected and, if necessary, the evaluation of the reintegration into the workplace is particularly relevant [52]. At this point the role of the occupational health unit healthcare personnel is essential to assess the impact of the disease on the workers and the effects on their functional capacity and adaptation at work [52].

In Spain, seeing the primary care physician and being referred to the ear, nose, and throat (ENT) specialist is the most common way for patients to try to find solutions. Seeking help from the primary care physician is a first option, and subsequent referral to a specialist (otorhinolaryngologist) is a common strategy for patients with olfactory loss and/or disorders [20]. In addition, some patients experienced social and financial difficulties due to the loss and/or alteration of smell and taste [5,20,51], because they could not afford to miss work to go to the doctor’s office or to undergo private rehabilitation treatment. The fact that many participants did not receive any treatment for the recovery of smell and/or taste is an experience reported in other studies such as Philpott et al. [20], Javed et al. [19], and Stankevice et al. [18]. It is worth highlighting how important it is for patients with Long-COVID-19 taste and smell disorders to be able to count on the involvement and concern of health professionals for improving their process. Sometimes, these symptoms are not even considered important by health professionals. Previous studies [8,18,19,27,53] showed how taste and smell impairments did not receive priority medical attention in the context of COVID-19, as such conditions were underestimated by health professionals. Burges Watson et al. [8] describe that a possible explanation for the lack of involvement of some professionals is a lack of empathy and a failure to understand the significance of symptom severity for patients. As a result, professionals underestimate the impact of symptoms, dismiss them, and cause patients to feel a sense of abandonment and to look outside the established care system for solutions [8]. We believe that the absence of external recognition by health professionals that they have a problem (Long-COVID-19) that requires follow-up, together with the absence of perception of severity and disability of the symptoms and the uncertainty in the face of changing symptoms (they come and go), influence help-seeking and can condition treatment and rehabilitation.

In our study there was a higher proportion of women. Previous studies showed that in the long term (the period between 180–360 days) the amount and presence of smell and taste alterations and other symptoms such as dyspnea did not differ between sexes [54] despite the fact that women have a higher risk of developing Long-COVID-19 [55,56]. Moreover, the study by Hirahata et al. [57] showed how women with Long-COVID-19, who had a reduced working day, were on sick leave, were fired or retired, or did not work, were associated with having a lower functional capacity. The consequences of infection in women could be related to gender and to the lack of studies focusing on the effect of COVID-19 on sexual and gender minorities [56].

### Strengths and Limitations

The strength of this study is that there are few qualitative studies describing the perspective of patients with Long-COVID-19 taste and smell loss and/or disorders [8,27]. The qualitative design enables us to explore and describe the participants’ perspectives in depth and helps us to understand olfactory and taste disorders for people with COVID-19 [8,22,23,24,25,27]. Compared to previous studies [8,27] on the perspective of patients with Long-COVID-19 with olfactory and taste disorders, our study adds these new findings: (a) these symptoms entail losing a gateway to their memories; (b) the alteration of taste does not always cause a change in negative dietary habits (more alcohol consumption, less consumption of vegetables), positive changes can also appear (more consumption of fish, less consumption of sweets); (c) people who have jobs related to smell and taste do not stop working because of the financial loss involved; (d) the possible consequences of the loss of taste are more tangible for the participants, but the symptom that most affects them is the loss of smell; (e) they eat with their sight, therefore the presentation of food (and identifying what they eat) becomes more important for these patients and facilitates intake; (f) the lack of updated information results in searching alternative sources such as the Internet without being able to verify the information; (g) the lack of a clear treatment leads to experimentation with all kinds of treatments, whether or not they are evidence-based; (h) professionals should avoid underestimating their symptoms, judging their decisions, or discouraging their attempts at finding a cure during the relationship with the patient.

The explanation for the differences compared to the studies by Almgren et al. and Burges Watson et al. [8,27] could be partly because in the case of Almgren et al., [27] their study focused on describing the patients’ perspective on Long-COVID-19. Although they found partial results on smell and taste symptoms, they did not focus specifically on these. Conversely, in the study by Burges Watson et al. [8], describing the perspective of Long-COVID-19 patients on smell and taste disorders, they used the COVID-19 Smell and Taste Loss Facebook group, where they posted a series of questions and received posts and responses from all users. This system could have limitations when it comes to delving into the individual experience and/or contrasting other information such as medical diagnosis and symptoms identified by professionals. Moreover, our study was conducted in a different social and health care setting.

The main limitation of the study, related to the own design, is that the results cannot be extrapolated to all post-COVID-19 patients with olfactory and/or taste disorders. Second, as for the suitability of the sample size, there is limited justification for this in qualitative health research, and defining the sample size a priori is problematic [39,40,58,59]. Different criteria may exist to finalize the data collection and/or saturate the information such as the following; theoretical saturation, inductive thematic saturation, a priori thematic saturation, sampling data saturation, and meaning saturation [39,40,58,59]. Nonetheless, we opted for an inductive thematic saturation proposal [39,40] where saturation focuses on the identification of new codes or themes rather than the completeness of existing theoretical categories. Previous studies [58,59] reported that in homogeneous groups of participants, with defined inclusion criteria shared by all participants, saturation can be reached with the completion of between 9–17 interviews. Finally, within the open interviews conducted, there may have been differences in the richness of the information provided and this could influence the results obtained.

## 5. Conclusions

Patients with Long-COVID-19 loss and/or disorders affecting smell and taste suffer changes and limitations in their habits (food and hygiene), in their daily life, and in their social relationships. In addition, when experiencing their symptoms, they may feel a lack of understanding from their environment and sometimes by professionals.

These findings have implications for practice by helping healthcare professionals to understand patients and assist them in their ability to cope with the consequences of the loss and/or disorders of taste and smell resulting from Long-COVID-19. The authors believe that within the evaluation and follow-up of patients with taste and smell symptoms due to Long-COVID-19, the nutritional and functional status should be monitored. This is due to the nutritional repercussions related to difficulties in the acquisition and choice of food, its preservation and preparation. Moreover, the symptoms could hinder functionality and autonomy by affecting basic activities of daily living (food preservation and preparation, lack or neglect of body hygiene), instrumental activities of daily living (buying and purchasing food and cooking), and advanced activities of daily living (establishing and continuing social relationships). Similarly, the psychological evaluation should study the effect of symptoms on self-confidence and self-esteem, and the presence of difficulties in establishing new social relationships (including intimate ones) or loss of previous ones. In addition to applying treatments, the role of professionals should include further interventions, such as the following: (a) providing contrasted and updated information on treatments and interventions with proven evidence-based effects; (b) monitoring the sources of information and unproven treatments used by their patients, and early detection of their (harmful) effects, in order to adopt strategies to control their symptoms; (c) teaching them to adopt strategies for remembering or controlling the quality of food, and encouraging their involvement in the purchase and preparation of their own meals, to increase their sense of control over their symptoms; (d) early detection of “false beliefs” about the unexpected “beneficial” effects of symptoms, such as weight loss. Burges Watson et al. [8] stated that weight loss was commonly reported among Long-COVID-19 patients with olfactory/taste disorders, however, this was not always considered to be a problem. Rather, it is thought that it may help certain individuals to begin adopting healthier habits.

## Figures and Tables

**Table 1 healthcare-12-00754-t001:** Semi-structured question guide.

Study Area	Question
Illness—follow-up	What has your experience of COVID-19 been like? What has been most relevant to you? What were your expectations about the disease and its evolution? What were your expectations for the follow-up of your symptoms when you left the hospital?
Daily life—strategies	How would you describe your alterations in taste and/or smell? How have these two symptoms affected your life? Have you made changes in your day-to-day life because of these two symptoms? How do you cope with them?
Family–social–work context	In relation to the alteration of taste and/or smell, has there been any impact on your family? Have you noticed changes in your social relationships, neighborhood, friends because of these two symptoms?
Treatment and recovery	Have you changed your hobbies because of these symptoms? Have you sought information about taste or smell disorders? Have you received any treatment for taste and/or smell disturbances? What expectations do you have for a cure for these two symptoms?

**Table 2 healthcare-12-00754-t002:** Trustworthiness criteria.

Criteria	Techniques Performed and Application Procedures
Credibility	Investigator triangulation: each interview was analyzed by two researchers. Thereafter, team meetings were performed in which the analyses were compared, and themes were identified. Triangulation of data collection methods: semi-structured interviews were conducted, and researcher field notes were kept. Member checking: the participants were asked to confirm the data obtained. All participants were offered the opportunity to review the audio and/or video records to confirm their experience. None of the participants made additional comments.
Transferability	In-depth descriptions of the study were performed, providing details of the characteristics of researchers, participants, contexts, sampling strategies, and the data collection and analysis procedures. Researchers were identified by their initials in the data collection section, which is a tool to ensure transferability in qualitative research.
Dependability	Audit by an external researcher: an external researcher assessed the study research protocol, focusing on aspects concerning the methods applied and study design.
Confirmability	Data collection and analysis triangulation. Researcher reflexivity was encouraged via the completion of reflexive reports and by describing the rationale for the study.

**Table 3 healthcare-12-00754-t003:** Clinical features of participants.

Other symptoms ^1^	Dyspnea at rest: *n* = 2 (16.7%) Exercise dyspnea: *n* = 7 (58.4%) Fatigue: *n* = 7 (58.4%) Muscle weakness: *n* = 3 (25%) Sleep disorders: *n* = 9 (75%) Muscle pain/loss of strength: *n* = 5 (41.6%) Hair loss: *n* = 2 (16.7%) Tachycardias/palpitations: *n* = 2 (16.7%) Rashes: *n* = 2 (16.7%) Memory loss: *n* = 5 (41.6%) Mental slowness: *n* = 2 (16.7%) Mental agitation/anxiety/fear: *n* = 10 (83.3%)
Post-COVID-19 symptoms (in addition to loss of taste and smell) ^1^	Two symptoms: *n* = 3 (25%) Three symptoms: *n* = 4 (33.3%) Four symptoms: *n* = 2 (16.7%) Five symptoms: *n* = 3 (25%)
Limitation of basic activities of daily living ^1,2^	None: *n* = 10 (83.3%) A little: *n* = 2 (16.7%)
Limitation of instrumental activities of daily living ^1,2^	None: *n* = 11 (91.7%) Moderate: *n* = 1 (8.3%)
Work limitation ^1,2^	Not at all: *n* = 7 (58.4%) A little: *n* = 3 (25%) Moderate: *n* = 1 (8.3%) Severe: *n* = 1 (8.3%)
Leisure limitation ^1,2^	None: *n* = 6 (50%) A little: *n* = 3 (25%) Moderate: *n* = 3 (25%)

^1^* n* represents the number of participants presenting other symptoms and/or limitations of activities of daily living, work, or leisure activities. ^2^ Evaluated using the Functional Impairment Checklist (FIC) [47,48].

**Table 4 healthcare-12-00754-t004:** Themes and categories.

Themes	Categories
Living with taste and smell disorders	Living with absence and/or altered sense of smell Living with absence and/or altered sense of taste The meaning of loss of smell and taste Lower-priority senses
Paths and challenges resulting from loss and/or impairment of taste and smell	Changes due to loss and/or alteration of the sense of smell Work-related difficulties due to loss of sense of smell Changes due to the loss and/or alteration of taste.
Coping with taste and smell disorders	Day-to-day strategies Health care

## Data Availability

The datasets generated and/or analyzed during the current study are not publicly available due to ethics restrictions but are available from the corresponding author on reasonable request.

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
