# Peer review of "Living and Coping with Olfactory and Taste Disorders: A Qualitative Study of People with Long-COVID-19"

_healthcare, 2024, doi:10.3390/healthcare12070754_

Round 1

Reviewer 1 Report

Comments and Suggestions for Authors

Dear authors, 

Thank you for let me reviewing the manuscript titled “Real-world experience of olfactory and taste disorders among people with post-COVID-19 condition: a qualitative study using in-depth interviews”.

I see as a qualitative study to drive attention about the olfactory and taste sequels in post-COVID 19 patients. I see it interesting to empathize with its limitations and undertand their situations.

Abstract

I suggest to include the sample and participants gender proportion.

Introduction

Just like an option, this reference, from the same authors country, could enrich the introduction and/or the discussion

Tíscar-González, V.; Sánchez-Gómez, S.; Martínez, L.; Serrano, P.; López, T.; Alonso, D.; Bartolomé-Rupérez, M.; Portuondo-Jiménez, J.; Zorrilla-Martínez, I. Vivencias e impacto en la calidad de vida de personas con COVID persistente. Gac. Sanit. 202337, 102247

Methods

Table 1 should be improved, in order that each study area were matched with its questions. Same comment to Table 2 about its ítems.

Results

Line 358, the word thyme is repeted twice

Discussion

I miss some discussion about the gender influence in the manuscript topic, specially considering that most of the participant were women. I suggest authors to work it.

Line 453, please define “ENT specialist”. About “Seeing the primary care physician and being referred to the ENT specialist is the most common way for patients to try to find solutions.”, please think about the occupational health specialist and unit. In fact, one person mentioned “…I specifically went to the occupational health department to report it…” (line 354). This workers have the right to be adapted their work. I suggest only to read the discussion of the following reference,

 https://doi.org/10.3390/healthcare11192632

In this sense I miss some practical performances that could be applied to cope with those limits.

Congratulations to authors

Author Response

Manuscript ID: healthcare-2891731
Type of manuscript: Article
Title: Real-world Experience of Olfactory and Taste Disorders among People with Post-COVID-19 Condition: A Qualitative Study Using In-Depth Interviews

We would like to thank the Editor and the Reviewers for their careful consideration of our manuscript. We would also like to thank the Reviewers for their suggestions, which we believe have enhanced the quality of the manuscript. We have highlighted all the changes we have made throughout the text in yellow. Below, please find a detailed list of how we have addressed each comment.

Dear authors,

Thank you for let me reviewing the manuscript titled “Real-world experience of olfactory and taste disorders among people with post-COVID-19 condition: a qualitative study using in-depth interviews”. I see as a qualitative study to drive attention about the olfactory and taste sequels in post-COVID 19 patients. I see it interesting to empathize with its limitations and undertand their situations.

Abstract

I suggest to include the sample and participants gender proportion.

RESPONSE:

We thank the reviewer for pointing this out. We have followed the reviewer´s suggestions, and the following new text has been included:

Twelve COVID-19 survivors (75% women) were recruited. The mean age of the participants was 55 years, and the mean duration of post-COVID symptoms was 25 months.

Introduction

Just like an option, this reference, from the same authors country, could enrich the introduction and/or the discussion. Tíscar-González, V.; Sánchez-Gómez, S.; Martínez, L.; Serrano, P.; López, T.; Alonso, D.; Bartolomé-Rupérez, M.; Portuondo-Jiménez, J.; Zorrilla-Martínez, I. Vivencias e impacto en la calidad de vida de personas con COVID persistente. Gac. Sanit. 2023, 37, 102247

RESPONSE:

Thank you for this suggestion, we have included this reference in the introduction section:

  • Tíscar-González V, Sánchez-Gómez S, Lafuente Martínez A, Peña Serrano A, Twose López M, Díaz Alonso S, Bartolomé-Rupérez M, Portuondo-Jiménez J, Zorrilla-Martínez I. Vivencias e impacto en la calidad de vida de personas con COVID persistente [Experiences and impact on the quality of life of people with long COVID]. Gac Sanit. 2023;37:102247. Spanish. doi: 10.1016/j.gaceta.2022.102247.

Methods

Table 1 should be improved, in order that each study area were matched with its questions.

RESPONSE:

We agree with the reviewer. However, the table was designed according to the journal’s guidelines.

For example, table in template of journal:

Table 1. This is a table. Tables should be placed in the main text near to the first time they are cited.

Title 1

Title 2

Title 3

entry 1

data

data

entry 2

data

data 1

1 Tables may have a footer.

There is NO line between entry 1 and entry 2. Similarly our table has no lines between the items/topics shown in the table.

Nonetheless, we have followed the reviewer’s suggestions and table 1 has been modified.

Same comment to Table 2 about its ítems.

RESPONSE:

We agree with the reviewer. However, the table was designed according to the journal’s guidelines.

Nonetheless, we have followed the reviewer’s suggestions and table 2 has been modified.

Results

Line 358, the word thyme is repeted twice

RESPONSE: Corrected.

Discussion

I miss some discussion about the gender influence in the manuscript topic, specially considering that most of the participant were women. I suggest authors to work it.

RESPONSE: We have followed the reviewer´s suggestions.

The discussion section has been edited as follows:

In our study there was a higher proportion of women. Previous studies show that in the long term (period between 180-360 days) the amount and presence of smell and taste alterations and other symptoms such as dyspnea do not differ between sexes [54] despite the fact that there women have a higher risk of developing Long COVID [55,56]. Moreover, the study by Hirahata et al. [57] showed how women with Long COVID, who had a reduced working day, were on sick leave, were fired or retired, or did not work, were associated with having a lower functional capacity. The consequences of infection in women could be related to gender and to the lack of studies focusing on the effect of COVID-19 on sexual and gender minorities [56].

New references have been added:

  • Mizrahi B, Sudry T, Flaks-Manov N, Yehezkelli Y, Kalkstein N, Akiva P, Ekka-Zohar A, Ben David SS, Lerner U, Bivas-Benita M, Greenfeld S. Long covid outcomes at one year after mild SARS-CoV-2 infection: nationwide cohort study. 2023 Jan 11;380:e072529. doi: 10.1136/bmj-2022-072529.
  • van Wijhe M, Fogh K, Ethelberg S, Karmark Iversen K, Nielsen H, Østergaard L, Andersen B, Bundgaard H, Jørgensen CS, Scharff BFS, Ellermann-Eriksen S, Johansen IS, Fomsgaard A, Grove Krause T, Wiese L, Fischer TK, Mølbak K, Benfield T, Folke F, Lippert F, Ostrowski SR, Koch A, Erikstrup C, Vangsted AM, Sørensen AIV, Ullum H, Skov RL, Simonsen L, Nielsen SD. Persistent Symptoms and Sequelae After Severe Acute Respiratory Syndrome Coronavirus 2 Infection Not Requiring Hospitalization: Results From Testing Denmark, a Danish Cross-sectional Survey. Open Forum Infect Dis. 2022 Dec 21;10(1):ofac679. doi: 10.1093/ofid/ofac679.
  • Laskowski NM, Brandt G, Paslakis G. Geschlechtsspezifische Unterschiede und Ungleichheiten der COVID-19 Pandemie: Eine Synthese systematischer Reviews unter Einbeziehung sexueller und geschlechtlicher Minderheiten [Gender Inequalities of the COVID-19 Pandemic: A Synthesis of Systematic Reviews with a Focus on Sexual and Gender Minorities]. Psychother Psychosom Med Psychol. 2024 Feb;74(2):57-69. German. doi: 10.1055/a-2228-6244.
  • Hirahata K, Nawa N, Fujiwara T. Characteristics of Long COVID: Cases from the First to the Fifth Wave in Greater Tokyo, Japan. J Clin Med. 2022 Oct 31;11(21):6457. doi: 10.3390/jcm11216457.

Line 453, please define “ENT specialist”. About “Seeing the primary care physician and being referred to the ENT specialist is the most common way for patients to try to find solutions.”.

RESPONSE: ENT refers to ear, nose, and throat (ENT) specialist

We have followed the Reviewer’s suggestion and this is now clarified in the text, see below:

Seeing the primary care physician and being referred to the ear, nose, and throat (ENT) specialist is the most common way for patients to try to find solutions.

, please think about the occupational health specialist and unit. In fact, one person mentioned “…I specifically went to the occupational health department to report it…” (line 354). This workers have the right to be adapted their work. I suggest only to read the discussion of the following reference, https://doi.org/10.3390/healthcare11192632

RESPONSE:

We have followed the reviewer´s suggestions.

Further information has been added concerning occupational health specialist, evaluation of COVID-19 consequences at work, and work adaptation at discussion section.

The following new text has been added:

In addition, the adaptation of work to the sequelae or symptoms of loss of smell and/or taste in those affected and, if necessary, the evaluation of the reintegration into the workplace is particularly relevant [52]. At this point the role of the occupational health units' healthcare personnel is essential to assess the impact of the disease on the workers and the effects on their functional capacity and adaptation at work [52].

Also, a new reference has been added:

  • Romero-Rodriguez E, Perula-de Torres LA, Monserrat-Villatoro J, Gonzalez-Lama J, Carmona-Casado AB, Ranchal-Sanchez A. Sociodemographic and Clinical Profile of Long COVID-19 Patients, and Its Correlation with Medical Leave: A Comprehensive Descriptive and Multicenter Study. Healthcare (Basel). 2023 Sep 27;11(19):2632. doi: 10.3390/healthcare11192632.

In this sense I miss some practical performances that could be applied to cope with those limits.

RESPONSE: We have followed the reviewer´s suggestions

We have now included practical implications to the conclusions section.

These findings have implications for practice by helping healthcare professionals to understand patients and assist them in their ability to cope with the consequences of the loss and/or disorders of taste and smell resulting from Long COVID. The authors believe that within the evaluation and follow-up of patients with taste and smell symptoms due to Long COVID, the nutritional and functional status should be monitored. This is due to the nutritional repercussions related to difficulties in the acquisition and choice of food, its preservation and preparation. Moreover, the symptoms could hinder functionality and autonomy by affecting basic activities of daily living (food preservation and preparation, lack or neglect of body hygiene), instrumental activities of daily living (buying and purchasing food and cooking) and advanced activities of daily living (establishing and continuing social relationships). Similarly, the psychological evaluation should study the effect of symptoms on self-confidence and self-esteem, and the presence of difficulties in establishing new social relationships (including intimate ones) or loss of previous ones. In addition to applying treatments, the role of professionals should include further interventions, such as: a) providing contrasted and updated information on treatments and interventions with proven evidence-based effects; b) monitoring the sources of information and unproven treatments used by their patients, and early detection of their (harmful) effects, in order to adopt strategies to control their symptoms; c) teaching them to adopt strategies for remembering or controlling the quality of food, and encouraging their involvement in the purchase and preparation of their own meals, to increase their sense of control over their symptoms; d) early detection of "false beliefs" about the unexpected "beneficial" effects of symptoms, such as weight loss. Burges Watson et al. [8] stated that weight loss was commonly reported among Long COVID patients with olfactory/taste disorders, however, this was not always considered to be a problem. Rather, it is thought that it may help certain individuals to begin adopting healthier habits.

Congratulations to authors

RESPONSE:

The authors thank the reviewer for this positive comment.

We hope that this revision is satisfactory and that the manuscript is now suitable for publication in Healthcare.

Sincerely,

The Authors

Reviewer 2 Report

Comments and Suggestions for Authors

Dear Authors,

First and foremost, allow me to commend your efforts in compiling and presenting the results of your research. It is a remarkable undertaking that contributes to our collective understanding of the subject. I would like to offer some constructive feedback aimed at improving the clarity, impact, and overall appeal of your manuscript to its intended audience.

Title

The current title of your study, while informative, does not fully capture the essence and novel contributions of your research. I suggest revising the title to more accurately reflect the innovative insights and findings that your work offers to the field.

Introduction

Adding a clear statement of the research questions and potential hypotheses toward the end of the Introduction section would significantly strengthen the reader's understanding of the study's objectives and expected results.

Section 2.1

On line 92 you have chosen a descriptive qualitative study. A rationale for this choice, explaining its alignment with the research objectives, would be beneficial. It would also be useful to add an argument as to how this design meets the aim of the research. It would also be useful to explain why a descriptive qualitative study was conducted when quantitative research (both descriptive and analytical) had already been published at the time of the study.

Section 2.2

The process for screening respondents needs to be explained. In addition, addressing how self-selection bias was mitigated would enhance the credibility of your study. The mention of COVID-19 survivors (line 110) should be accompanied by specific participant numbers, and the basis for identifying olfactory and taste symptoms-whether self-reported or clinically assessed-needs clarification. Please include this information in the appropriate section (presumably Section 2.3).

Section 2.3

Abbreviations are given on lines 132 and 133; although it is clear that you have shared the authors' initials, it would be useful to explain the abbreviations in the final text.

Conclusions section

The inclusion of details about the course of COVID-19 among the participants, including hospitalization, vaccination status, and which particular variant they were infected with, would enrich the conclusions. Highlighting the spectrum of other COVID-19 symptoms experienced would also be valuable.

Results section

The current over-reliance on citations detracts from the original contributions of your study. The results section contains an excessive number of citations, but in this form they are of little informative value. The accompanying comments, if any, merely repeat these observations. However, a significant number of excerpts from the interviews conducted are not mentioned at all in the text, and it is therefore not clear in what way the authors consider the citation to be relevant.

Discussion section

The nature of the text is more like an introduction or literature review. Therefore, please include a real discussion of the results obtained. Acknowledging previous studies while clearly articulating how your research advances the field is essential.

Concluding Remarks

The conclusions drawn seem somewhat cursory. Each statement in this section should be robustly supported by empirical evidence from your study, rather than simply reiterating common knowledge.

General Recommendations

It is critical that the revised manuscript clearly delineate the novel contributions of your research and articulate the unique insights and previously unrecognized knowledge that it brings to the forefront. In addition, the manuscript would benefit from a thorough analysis and synthesis of the collected data, addressing perceived challenges, coping mechanisms, and other important aspects that have not been previously explored.

I trust that these suggestions will serve as valuable guidance in refining your manuscript. Your commitment to advancing our understanding of this topic is greatly appreciated, and I look forward to seeing the improved version of your paper.

Thank you for your attention to these recommendations, and I wish you all the best in your current and future research endeavors.

Sincerely,

Author Response

Manuscript ID: healthcare-2891731
Type of manuscript: Article
Title: Real-world Experience of Olfactory and Taste Disorders among People with Post-COVID-19 Condition: A Qualitative Study Using In-Depth Interviews

We would like to thank the Editor and the Reviewers for their careful consideration of our manuscript. We would also like to thank the Reviewers for their suggestions, which we believe have enhanced the quality of the manuscript. We have highlighted all the changes we have made throughout the text in yellow. Below, please find a detailed list of how we have addressed each comment.

Dear Authors,

First and foremost, allow me to commend your efforts in compiling and presenting the results of your research. It is a remarkable undertaking that contributes to our collective understanding of the subject. I would like to offer some constructive feedback aimed at improving the clarity, impact, and overall appeal of your manuscript to its intended audience.

Title

The current title of your study, while informative, does not fully capture the essence and novel contributions of your research. I suggest revising the title to more accurately reflect the innovative insights and findings that your work offers to the field.

RESPONSE:

Due to the number of results, it is not possible to include all of these contributions in the title; the new insights are included in two terms that the authors believe reflect all the results of the study (living and coping), and therefore the title has been edited as follows on the basis of the reviewer's indications.

Living and coping with Olfactory and Taste Disorders: A qualitative study of people with Long COVID.

Introduction

Adding a clear statement of the research questions and potential hypotheses toward the end of the Introduction section would significantly strengthen the reader's understanding of the study's objectives and expected results.

RESPONSE:

We have followed the reviewer’s suggestion.

We have now included research questions following the EPPIC framework (Luciani et al., 2019). The EPPIC proposal meaning the Emphasis-Purposeful sample-Phenomenon of interest-Context framework (Jack & Phoenix, 2022).

We have followed the suggestions by Jack & Phoenix (2022) regarding “Research problem and purpose statement” (page 832) and “Overarching research question” (page 832-833). We have used the Emphasis-Purposeful sample-Phenomenon of interest-Context framework (EPPiC) to build the research question.

Jack & Phoenix (2022) regarding “Research problem and purpose statement” reported: “In QHR [qualitative health research], core components of a purpose statement would include: (1) the research design; (2) the qualitative function of the study (e.g. description, exploration, development, explanation); and (3) identification of the phenomenon of interest being studied.” (page 832)

The same author regarding “Overarching research question” reported that: “In QHR, application of the Emphasis-Purposeful sample-Phenomenon of interest-Context framework (EPPiC) can assist researchers in writing their research question by including four critical components exemplified by language that: (1) infers the study’s purpose or emphasis; (2) describes the purposeful sample of participants; (3) identifies the health phenomenon of interest being described, explored, evaluated, or explained; and (4) indicates the context that shapes participants’ experiences and perceptions of the phenomena of interest.” (pages 832-833).

Also, Moisey et al. (2022) reported how: “Using the EPPiC framework, the four core components of a structured research question include: (1) language that reflects the purpose, function or (E) emphasis of the study; (2) descriptive language describing the (P) purposeful sample of participants; (3) identification of the specific social or human (Pi) phenomenon of interest being described, explored, evaluated or explained; and (4) identification of the (C) context that shapes participants’ experiences and perceptions of the phenomena of interest.”(page 378)

The introduction section has been edited to include new text as follows:

However, further studies are needed to describe the experience of living with long-lasting loss of taste and/or smell among people with Long-COVID, the consequences of these disorders and how patients cope and develop strategies to deal with them in different geographical and cultural contexts. The guiding question for this study, based on the EPPIC framework, was [28-30]: What is the perspective of people with Long-COVID with taste and smell disorders who belong to Spanish Long-COVID patient associations? In addition, how was the daily life of these people? What challenges and obstacles did they face? How did they deal with these difficulties? The aim of this study was to describe the perspective of a group of individuals with Long-COVID taste and smell disorders regarding their daily life.

References:

  • Moisey LL, A Campbell K, Whitmore C, Jack SM. Advancing qualitative health research approaches in applied nutrition research. J Hum Nutr Diet. 2022 Apr;35(2):376-387. doi: 10.1111/jhn.12989. Epub 2022 Jan 23. PMID: 34997658.
  • Luciani M, Campbell K, Tschirhart H, Ausili D, Jack SM. How to design a qualitative health research study. Part 1: design and purposeful sampling considerations. Prof Inferm. 2019;72(2):152–161.
  • Jack SM, Phoenix Qualitative health research in the fields of developmental medicine and child neurology. Dev Med Child Neurol. 2022 Jul;64(7):830-839. doi: 10.1111/dmcn.15182. Epub 2022 Feb 13. PMID: 35156198.

 Section 2.1

On line 92 you have chosen a descriptive qualitative study. A rationale for this choice, explaining its alignment with the research objectives, would be beneficial. It would also be useful to add an argument as to how this design meets the aim of the research. It would also be useful to explain why a descriptive qualitative study was conducted when quantitative research (both descriptive and analytical) had already been published at the time of the study.

RESPONSE:

The research question determines the methodology to be used to answer the question. In the present study, the question established following the EPPiC proposal is oriented to studying people's perspective and experience. Therefore, qualitative methodology should be used.

As Herbert et al. (2011) pointed out in their work on clinical evidence in health sciences, clinical evidence is underpinned by asking relevant clinical questions. These questions are oriented towards effects of intervention, patients’ experiences, the course of a condition (prognosis), and the accuracy of diagnostic tests (page 10).

In the case of patients' experiences, there are various techniques and methodologies that can help to answer them, such as clinical observation and qualitative studies. In the case of patients' experiences, qualitative research is recommended.

Herbert et al. (2011) reported that: “Questions about experiences are best answered by qualitative methods. Qualitative research methods, also called methods of naturalistic inquiry, were developed in the social and human sciences and refer to theories on interpretation (hermeneutics) and human experience (phenomenology) (…) are useful for the study of human and social experience, communication, thoughts, expectations, meanings, attitudes and processes, especially those related to interaction, relations, development, interpretation, movement and activity…” (page 27).

Within qualitative methodology, there are numerous proposals and types of design. To select the appropriate design, it should be based on the research question (Jack & Phoenix, 2022; Moisey et al., 2022).

In this case the question was: What is the perspective of people with Long-COVID with taste and smell disorders who belong to Spanish Long-COVID patient associations? In addition, how was the daily life of these people? What challenges and obstacles did they face? How did they deal with these difficulties?

Following the EPPiC proposal recommendations: “Using the EPPiC framework, the four core components of a structured research question include: (1) language that reflects the purpose, function or (E) emphasis of the study; (2) descriptive language describing the (P) purposeful sample of participants; (3) identification of the specific social or human (Pi) phenomenon of interest being described, explored, evaluated or explained; and (4) identification of the (C) context that shapes participants’ experiences and perceptions of the phenomena of interest.”(Moisey et al., 2022: page 378).

This means that the way of establishing the question and the use of language (What, how, why) determines the type of qualitative design to choose.

Jack & Phoenix (2022) and Moisley et al. (2022) describe how in health sciences, there are different types of qualitative designs that can be used such as: Qualitative description, interpretive description, Focused ethnography, Descriptive phenomenology, Interpretive phenomenology, and Traditional grounded theory.

The reporting of this research and the descriptions of the research question(s) are in the form of a descriptive qualitative study.

Jack & Phoenix (2022) showed that a qualitative descriptive study based on Qualitative description using this qualitative health research study design should be used when the purpose of the project is to: a) Describe the phenomenon of interest or the perspective of the people who experience it, b) Identify individual, relational, team, unit/organizational, community (or social) factors that serve as barriers/facilitators with respect to a specific health issue or action, and c) Understand the nature of a health practice, policy, or education ‘problem’, challenge, or issue. (page 835, table 3).

Also, Sandelowski (2000; 2010) reported that qualitative descriptive studies are aimed at providing a comprehensive summary of events in the everyday terms of those events. Researchers conducting qualitative descriptive studies stay close to their data and to the surface of words and events. Moreover, qualitative descriptive designs are typically an eclectic but reasonable combination of sampling, and data collection, analysis, and re-presentation techniques. Finally, a qualitative descriptive study is the method of choice when straight descriptions of phenomena are desired. Thus, considering the chosen design, this justifies the data collection method (in-depth interviews), and the type of analysis used (inductive thematic analysis).

Further information has been added to the manuscript to explain the qualitative design:

A descriptive qualitative study, based on an interpretative paradigm [28,31-33] according to the Standards for Reporting Qualitative Research [34] and the Consolidated Criteria for Reporting Qualitative Research [35] was conducted. Descriptive qualitative designs are based on the participants' own words to describe their personal experience, as well as their perspectives on certain phenomena such as illness and disability [31,36,37]. The aim of a descriptive qualitative study is to identify an event (disease, health problem, symptoms, condition) or a critical situation [28,31,32]. This type of qualitative design enables the use of a wide range of data collection instruments, and different proposals for analysis, to describe what is happening and how is it happening from the perspective of those involved in their own environment [32,33]. Qualitative descriptive studies aim to be a comprehensive summary of events in the everyday terms of the described event [33,36,37]. This design is the method of choice when straight descriptions of phenomena are desired [33,36,37].

Also, we have included new references:

  • Sandelowski M, Barroso J. Classifying the findings in qualitative studies. Qual Health Res. 2003;13(7):905-23. doi: 10.1177/1049732303253488.
  • Sandelowski M. Whatever happened to qualitative description? Res Nurs Health. 2000;23(4):334-40. doi: 10.1002/1098-240x(200008)23:4<334::aid-nur9>3.0.co;2-g.
  • Sandelowski M. What's in a name? Qualitative description revisited. Res Nurs Health. 2010;33(1):77-84. doi: 10.1002/nur.20362.

References:

  • Herbert R, Jamtvedt G, Birger Hagen K, Mead J. Practical Evidence-Based Physiotherapy. 2 ed. Elsevier, 2011.
  • Jack SM, Phoenix M. Qualitative health research in the fields of developmental medicine and child neurology. Dev Med Child Neurol. 2022 Jul;64(7):830-839. doi: 10.1111/dmcn.15182.
  • Moisey LL, A Campbell K, Whitmore C, Jack SM. Advancing qualitative health research approaches in applied nutrition research. J Hum Nutr Diet. 2022 Apr;35(2):376-387. doi: 10.1111/jhn.12989. Epub 2022 Jan 23. PMID: 34997658.

Concerning the following sentence by the reviewer: “It would also be useful to explain why a descriptive qualitative study was conducted when quantitative research (both descriptive and analytical) had already been published at the time of the study.”

RESPONSE:

Certainly, there is a large number of published studies using quantitative methodology on covid-19, long covid and loss of smell and taste due to the virus. However, this is not so for studies based on qualitative methodology.

The authors are aware that there is currently insufficient evidence from qualitative studies on the phenomenon of altered taste and smell in people with Long COVID (see justification of evidence with data below). This justifies the need for further study of this phenomenon. Specifically, this study shows the perspective of a group of Long COVID patients, who belong to the only two Long COVID associations in Spain with taste and smell disorders. This justifies a very specific context in a given circumstance.

Performing a search in pub med/med line on this phenomenon, including the keywords, qualitative research and COVId-19 (thesaurus terms at MeSH) and taste and los term (entry terms at MeSH), the following results were obtained that justified the present study:

Pub med search, March 4, 2024. https://pubmed.ncbi.nlm.nih.gov/?term=%28COVID-19+AND+QUALITATIVE+RESEARCH+AND+taste+loss%29&sort=pubdate&size=50

(COVID-19 AND QUALITATIVE RESEARCH AND taste loss)

Nineteen results were obtained. Of these, 6 used qualitative methodology, and 4 were related to the objective of the study. Of these 4, only one was on the perspective of symptoms in patients with long-covid (nº15, doi: 10.1371/journal.pone.0256998)

  1. doi: 10.4103/IJO.IJO_1437_23.
  2. doi: 10.7759/cureus.46506.
  3. doi: 10.1371/journal.pone.0290693.
  4. doi: 10.1002/alr.23100.
  5. doi: 10.1002/hsr2.1400.
  6. doi: 10.3390/v15071418.
  7. doi: 10.1007/s11136-022-03336-3.
  8. doi: 10.1016/j.nutos.2022.11.008. It used qualitative research (QR), and include the topic under study
  9. doi: 10.1002/pds.5564. It used QR, but not the topic under study
  10. doi: 10.3390/ijerph192113980. It used QR, but not the topic under study
  11. doi: 10.4193/Rhin21.415.
  12. doi: 10.1136/bmjopen-2021-055989. It used QR, and include the topic under study
  13. doi: 10.3390/life12030461.
  14. doi: 10.3390/foods11040607. It used QR, and include the topic under study
  15. doi: 10.1371/journal.pone.0256998. It used QR, and include the topic under study
  16. doi: 10.1007/s00259-021-05294-3.
  17. doi: 10.1136/bmjopen-2020-046276.
  18. doi: 10.1093/chemse/bjaa041.
  19. doi: 10.1212/WNL.0000000000009619.

Pub med search, March 4, 2024. https://pubmed.ncbi.nlm.nih.gov/?term=%28COVID-19+AND+QUALITATIVE+RESEARCH+AND+smell+loss%29&sort=pubdate&size=50

(COVID-19 AND QUALITATIVE RESEARCH AND smell loss)

33 results. Of these, 9 were qualitative, and of these, 5 were related to the topic under study. Of these 5, 3 were repeated (#19, #21, and #23). There were only two new articles (#16 and #22). And of those two new articles, only one was about longcovid (nº16- doi: 10.3390/ijerph192113980.)

  1. doi: 10.4103/IJO.IJO_1437_23.
  2. doi: 10.3390/nu15214538.
  3. doi: 10.7759/cureus.46506.
  4. doi: 10.1371/journal.pone.0290693. It used QR, but not the topic under study
  5. doi: 10.1002/ohn.352.
  6. doi: 10.1002/alr.23100.
  7. DOI: 10.1002/hsr2.1400
  8. doi: 10.3390/v15071418.
  9. doi: 10.3389/fpsyg.2023.1190994.
  10. doi: 10.1007/s11136-022-03336-3.
  11. doi: 10.1016/j.nutos.2022.11.008.
  12. doi: 10.1002/pds.5564. REPEATED. It used QR, but not the topic under study
  13. doi: 10.1186/s41687-022-00535-x. It used QR, but not the topic under study
  14. doi: 10.1093/chemse/bjad002.
  15. doi: 10.1177/19458924221120117.
  16. doi: 10.3390/ijerph192113980. It used QR, and include the topic under study
  17. doi: 10.33594/000000531.
  18. doi: 10.4193/Rhin21.415.
  19. doi: 10.1136/bmjopen-2021-055989. REPEATED. It used QR, and include the topic under study
  20. doi: 10.3390/life12030461.
  21. doi: 10.3390/foods11040607. It used QR, and include the topic under study
  22. doi: 10.2196/29086. It used QR, and include the topic under study
  23. doi: 10.1371/journal.pone.0256998. REPEATED. It used QR, and include the topic under study
  24. PMID: 34559791.
  25. doi: 10.14814/phy2.14992.
  26. doi: 10.1007/s00259-021-05294-3.
  27. doi: 10.3389/fpubh.2021.716421. It used QR, but not the topic under study
  28. doi: 10.1136/bmjopen-2020-046276.
  29. doi: 10.1016/j.acra.2020.10.006.
  30. doi: 10.1093/chemse/bjaa041.
  31. doi: 10.1016/j.semarthrit.2020.06.012.
  32. doi: 10.1212/WNL.0000000000009619.
  33. doi: 10.3389/fneur.2020.00687.

In total between the 2 searches, 6 studies were identified with qualitative methodology and within the topic under study, and only 2 qualitative studies of the topic under study in patients with long covid (doi: 10.1371/journal.pone.0256998/  doi: 10.3390/ijerph192113980).

These results have been included in the present work:

  • Almgren J, Löfström E, Malmborg JS, Nygren J, Undén J, Larsson I. Patients' Health Experiences of Post COVID-19 Condition-A Qualitative Study. Int J Environ Res Public Health. 2022 Oct 27;19(21):13980. doi: 10.3390/ijerph192113980. PMID: 36360860; PMCID: PMC9656359.
  • Burges Watson DL, Campbell M, Hopkins C, Smith B, Kelly C, Deary V. Altered smell and taste: Anosmia, parosmia and the impact of long Covid-19. PLoS One. 2021 Sep 24;16(9):e0256998. doi: 10.1371/journal.pone.0256998. PMID: 34559820; PMCID: PMC8462678.
  • Kelly CE. Parosmia and altered taste in patients recovering from Covid 19. Clin Nutr Open Sci. 2023 Apr;48:1-10. doi: 10.1016/j.nutos.2022.11.008.
  • Parker JK, Kelly CE, Smith BC, Kirkwood AF, Hopkins C, Gane S. Patients' Perspectives on Qualitative Olfactory Dysfunction: Thematic Analysis of Social Media Posts. JMIR Form Res. 2021 Dec 14;5(12):e29086. doi: 10.2196/29086.
  • Rofail D, McGale N, Podolanczuk AJ, Rams A, Przydzial K, Sivapalasingam S, Mastey V, Marquis P. Patient experience of symptoms and impacts of COVID-19: a qualitative investigation with symptomatic outpatients. BMJ Open. 2022 May 2;12(5):e055989. doi: 10.1136/bmjopen-2021-055989.
  • Turner L, Rogers PJ. Varied Effects of COVID-19 Chemosensory Loss and Distortion on Appetite: Implications for Understanding Motives for Eating and Drinking. Foods. 2022 Feb 20;11(4):607. doi: 10.3390/foods11040607.

The authors believe that the publication of 6 articles on the topic, using qualitative methods that describe and analyze the loss of smell and taste, is not sufficient to describe the phenomenon in depth in all its dimensions and it is necessary to continue studying the phenomenon in different geographical and cultural contexts, which is why the present work is justified. Moreover, there are specifically only two articles that have been carried out on patients with Long Covid. One of these is the study by Burges Watson et al (2021) in the UK, and the other is the study by Almgren et al (2022) in Sweden.

  • (Sweden) Almgren J, Löfström E, Malmborg JS, Nygren J, Undén J, Larsson I. Patients' Health Experiences of Post COVID-19 Condition-A Qualitative Study. Int J Environ Res Public Health. 2022 Oct 27;19(21):13980. doi: 10.3390/ijerph192113980. PMID: 36360860; PMCID: PMC9656359.
  • (UK) Burges Watson DL, Campbell M, Hopkins C, Smith B, Kelly C, Deary V. Altered smell and taste: Anosmia, parosmia and the impact of long Covid-19. PLoS One. 2021 Sep 24;16(9):e0256998. doi: 10.1371/journal.pone.0256998. PMID: 34559820; PMCID: PMC8462678.

The chronic nature of a disease is a priority in studies with qualitative methodology due to the multidimensional nature that can influence the experience, not only of the sick person, but also of his or her family, social and work environment (Shelton et al., 2022). Even Burges Watson et al. (2021) in their final recommendations suggest continuing qualitative studies on the alteration and/or loss of taste and smell in Long-COVID. Thus, it is necessary to continue developing studies on the perspective of patients with symptomatology that remains (with Long-COVID) after infection with SARS-CoV-2 virus.

Also, we have redrafted part of the introduction, including new information, to improve the final text. We included:

Previous observational studies have demonstrated that people who suffer from post-COVID taste and smell disorders suffer a decline in their daily and social quality of life, affecting their eating habits and customs, and negatively impacting their physical and mental wellbeing [9,21]. Furthermore, there are few qualitative studies [22-25] on the experience of patients describing their loss and/or alterations of taste and smell due to SARS-CoV-2 acute infection. The available studies [22-25] report how this condition has decreased their ability to detect hazards such as toxic odors, smoke/fire or gas leaks is decreased. Furthermore, distortions of smell have a direct impact on the experience of eating and enjoyment of food since the taste of food is the result of the combination of smell and taste [22,23,25].

The presence of symptoms that are perpetuated over time and can become chronic conditions the perspective of patients and their families. Also, it influences their expectations of cure, adherence to treatment and the development of strategies to enable them to adapt to the symptoms and changes caused by the disease [26]. There is limited literature available to date describing the patients' perspective and experience of Long-COVID loss of taste and smell [8,27]. Almgren et al. [27] (Sweden) and Burges Watson et al. [8] (UK) reported how olfactory and taste dysfunctions in patients with Long-COVID decreased self-confidence, leading to a decreased desire to eat and impaired ability to prepare food, changes in body weight, which may even affect patients’ self-perception and self-esteem, leading to an altered relationship to oneself and others.

However, further studies are needed to describe the experience of living with long-lasting loss of taste and/or smell among people with Long-COVID, the consequences of these disorders and how patients cope and develop strategies to deal with them in different geographical and cultural contexts. The guiding question for this study, based on the EPPIC framework, was [28-30]: What is the perspective of people with Long-COVID with taste and smell disorders who belong to Spanish Long-COVID patient associations? In addition, how was the daily life of these people? What challenges and obstacles did they face? How did they deal with these difficulties? The aim of this study was to describe the perspective of a group of individuals with Long-COVID taste and smell disorders regarding their daily life.

References:

  • Shelton RC, Philbin MM, Ramanadhan S. Qualitative Research Methods in Chronic Disease: Introduction and Opportunities to Promote Health Equity. Annu Rev Public Health. 2022 Apr 5;43:37-57. doi: 10.1146/annurev-publhealth-012420-105104.

Section 2.2

The process for screening respondents needs to be explained. In addition, addressing how self-selection bias was mitigated would enhance the credibility of your study. The mention of COVID-19 survivors (line 110) should be accompanied by specific participant numbers, and the basis for identifying olfactory and taste symptoms-whether self-reported or clinically assessed-needs clarification. Please include this information in the appropriate section (presumably Section 2.3).

RESPONSE:

The participants were recruited from two Spanish associations of people affected with Long Covid. In qualitative research, sampling strategies are used to recruit participants who have relevant information to answer the research question; it is not considered a bias. This is a customary approach used in qualitative research.

Moser & Korstjens (2018) reported that: “The key features of a qualitative sampling plan are as follows. First, participants are always sampled deliberately. Second, sample size differs for each study and is small. Third, the sample will emerge during the study: based on further questions raised in the process of data collection and analysis, inclusion and exclusion criteria might be altered, or the sampling sites might be changed. Finally, the sample is determined by conceptual requirements and not primarily by representativeness (…) The sampling plan is appropriate when the selected participants and settings are sufficient to provide the information needed for a full understanding of the phenomenon under study (…) Subsequently, the best strategy to apply is to recruit participants who can provide the richest information (…) Sampling is the process of selecting or searching for situations, context and/or participants who provide rich data of the phenomenon of interest. In qualitative research, you sample deliberately, not at random. The most commonly used deliberate sampling strategies are purposive sampling, criterion sampling, theoretical sampling, convenience sampling and snowball sampling (…) Key informants hold special and expert knowledge about the phenomenon to be studied and are willing to share information and insights with you as the researcher.” (page 10).

Also, Klem et al (2022) described that: “Sampling focuses on recruiting a specific group of people who have experienced a phenomenon of interest. Sampling strategies will vary according to the research question (…) you will often see purposive sampling, an intentional selection of informants based on their ability to elucidate a specific theme, concept, or phenomenon.” (page 8-9)

Finally, Jack & Phoenix (2022) reported that: “QHR [qualitative health research] most often includes purposeful, sometimes referred to as purposive, sampling methods in which the participants are chosen according to their specific identity and experiences of or with the phenomenon of interest under study. People who take part in qualitative research are typically referred to as participants, not subjects, and they are not expected to represent a larger or normative population. Instead, participants may be chosen for having a unique and often more difficult to access perspective (e.g. populations who typically face barriers to participating in health services and research).” (page 835)

On the other hand, credibility was verified using the following procedures: a) Investigator triangulation: each interview was analyzed by two researchers. Thereafter, team meetings were performed in which the analyses were compared, and themes were identified; b) Triangulation of data collection methods: semi-structured interviews were conducted, and researcher field notes were kept; and c) Member checking: the participants were asked to confirm the data obtained. All participants were offered the opportunity to review the audio and/or video records to confirm their experience. None of the participants made additional comments.

See table 2. Trustworthiness criteria.

On the other hand, the researchers do not know the total number of people with long covid in Spain, or in the associations, because participants were included from the associations through flyers, social networks, and internet platforms for COVID-19 support groups. We could not know the exact number of participants in each social network or support group. Participants (if they met the inclusion criteria) were included consecutively in the study, in a progressive manner.

Taste and smell symptoms were confirmed by medical specialists (otorhinolaryngologist).

All observations and changes made following the reviewer's indications are included in section 2.2. Participants, context, and sampling strategies.

Purposive sampling was applied to select participants with relevant information for the study [38]. This type of sampling strategy focuses on deliberately recruiting a specific group of people who have experienced a phenomenon of interest [38]. Thus, participants were recruited from two COVID-19 patient organizations (Long COVID ACTS and COVID persistente España [Persistent COVID Spain]) through flyers, social networks, and internet platforms for COVID-19 support groups. All participants from the two associations who voluntarily wished to participate were included consecutively. The inclusion criteria consisted of COVID-19 survivors, aged over 18 years with long-lasting post-COVID olfactory and taste symptoms (total or partial loss) (confirmed by medical specialist, otorhinolaryngologist) for at least three months duration after COVID-19 diagnosis, as confirmed by positive reverse-transcription-polymerase chain reaction test from a nasopharyngeal or oropharyngeal swab.

There was a mistake in the manuscript, since section 2.3 was about data collection and it was repeated that it was about participants. All this new information is included in section 2.2 Participants.

References:

  • Jack SM, Phoenix M. Qualitative health research in the fields of developmental medicine and child neurology. Dev Med Child Neurol. 2022 Jul;64(7):830-839. doi: 10.1111/dmcn.15182.
  • Klem NR, Shields N, Smith A, Bunzli S. Demystifying Qualitative Research for Musculoskeletal Practitioners Part 4: A Qualitative Researcher's Toolkit-Sampling, Data Collection Methods, and Data Analysis. J Orthop Sports Phys Ther. 2022 Jan;52(1):8-10. doi: 10.2519/jospt.2022.10486.
  • Moser A, Korstjens I. Series: Practical guidance to qualitative research. Part 3: Sampling, data collection and analysis. Eur J Gen Pract. 2018 Dec;24(1):9-18. doi: 10.1080/13814788.2017.1375091.

Section 2.3

Abbreviations are given on lines 132 and 133; although it is clear that you have shared the authors' initials, it would be useful to explain the abbreviations in the final text.

 RESPONSE:

We agree with the reviewer.

We have edited the rigor section as follows (table 2):

Transferability

In-depth descriptions of the study were performed, providing details of the characteristics of researchers, participants, contexts, sampling strategies, and the data collection and analysis procedures. Researchers are identified by their initials in the data collection section, which is a tool to ensure transferability in in qualitative research.

Conclusions section

The inclusion of details about the course of COVID-19 among the participants, including hospitalization, vaccination status, and which particular variant they were infected with, would enrich the conclusions. Highlighting the spectrum of other COVID-19 symptoms experienced would also be valuable.

 RESPONSE:

The authors do not have all the new data suggested by the reviewer. However, some are presented in a new Table 3. The new table is included in the results section, see below:

Table 3. Clinical features of participants.

Other symptoms 1

Dyspnea at rest: n=2 (16.66%)

Exercise dyspnea: n= 7 (58.33%)

Fatigue: n=7 (58.33%)

Muscle weakness: n=3 (25%)

Sleep disorders: n=9 (75%)

Muscle pain/loss of strength: n=5 (41.6%)

Hair loss: n=2 (16.66%)

Tachycardias/palpitations: n=2 (16.66%)

Rashes: n=2 (16.66%)

Memory loss: n=5 (41.6%)

Mental slowness: n=2 (16.66%)

Mental agitation/anxiety/fear: n=10 (83.3%)

Post-Covid symptoms (in addition to loss of taste and smell)1

Two symptoms: n=3 (25%)

Three symptoms: n=4 (33.3%)

Four symptoms: n=2 (16.6%)

Five symptoms: n=3 (25%)

Limitation of basic activities of daily living 1,2

None: n=10 (83.3%)

A little: n=2 (16.6%)

Limitation of instrumental activities of daily living 1,2

None: n=11 (91.6%)

Moderate: n=1 (8.3%)

Work limitation 1,2

Not at all: n=7 (58.3%)

A little: n=3 (25%)

Moderate: n=1 (8.3%)

Severe: n=1 (8.3%)

Leisure limitation 1,2

None: n= 6 (50%)

A little: n=3 (25%)

Moderate: n=3 (25%)

1 n represents the number of participants presenting other symptoms and/or limitations of activities of daily living, work, or leisure activities.2 Evaluated using the Functional Impairment Checklist (FIC) [47,48]

We have also included new references:

  • Fernández-de-Las-Peñas C, Palacios-Ceña M, Rodríguez-Jiménez J, de-la-Llave-Rincón AI, Fuensalida-Novo S, Cigarán-Méndez M, Florencio LL, Ambite-Quesada S, Ortega-Santiago R, Pardo-Hernández A, Hernández-Barrera V, Palacios-Ceña D, Gil-de-Miguel Á. Psychometric Properties of the Functional Impairment Checklist (FIC) as a Disease-Specific Patient-Reported Outcome Measure (PROM) in Previously Hospitalized COVID-19 Survivors with Long-COVID. Int J Environ Res Public Health. 2022 Sep 12;19(18):11460. doi: 10.3390/ijerph191811460.
  • Lam SP, Tsui E, Chan KS, Lam CL, So HP. The validity and reliability of the functional impairment checklist (FIC) in the evaluation of functional consequences of severe acute respiratory distress syndrome (SARS). Qual Life Res. 2006 Mar;15(2):217-31. doi: 10.1007/s11136-005-1463-5.

Results section: The current over-reliance on citations detracts from the original contributions of your study. The results section contains an excessive number of citations, but in this form they are of little informative value. The accompanying comments, if any, merely repeat these observations. However, a significant number of excerpts from the interviews conducted are not mentioned at all in the text, and it is therefore not clear in what way the authors consider the citation to be relevant.

Discussion section: The nature of the text is more like an introduction or literature review. Therefore, please include a real discussion of the results obtained. Acknowledging previous studies while clearly articulating how your research advances the field is essential.

RESPONSE:

Thank you for your insight, we have followed your suggestions to further improve our manuscript.

We have rewritten the results and discussion section. In the results we have revised the number of quotes, we have provided more explanatory information on the findings identified by the researchers in order to show greater coherence between the explanation and the examples of narratives used. Furthermore, in the discussion we have included similarities and differences with other studies directly linked to the results, incorporated possible explanations for these similarities or differences, and added explanations of the implications of our results for our field of research. Practical implications have been detailed, as well as novel findings identified in our study compared to previous studies.

The authors believe that this will improve clarity and improve the presentation and synthesis of the results and their subsequent discussion.

The changes highlighted in yellow can be seen in the results and discussion section.

Concluding Remarks

The conclusions drawn seem somewhat cursory. Each statement in this section should be robustly supported by empirical evidence from your study, rather than simply reiterating common knowledge.

RESPONSE: We have made edits to this section, as suggested by the reviewer, new text is now included:

Patients with Long COVID loss and/or disorders affecting smell and taste suffer changes and limitations in their habits (food and hygiene), in their daily life, and in their social relationships. In addition, when experiencing their symptoms, they may feel a lack of understanding by their environment and sometimes by professionals.

These findings have implications for practice by helping healthcare professionals to understand patients and assist them in their ability to cope with the consequences of the loss and/or disorders of taste and smell resulting from Long COVID. The authors believe that within the evaluation and follow-up of patients with taste and smell symptoms due to Long COVID, the nutritional and functional status should be monitored. This is due to the nutritional repercussions related to difficulties in the acquisition and choice of food, its preservation and preparation. Moreover, the symptoms could hinder functionality and autonomy by affecting basic activities of daily living (food preservation and preparation, lack or neglect of body hygiene), instrumental activities of daily living (buying and purchasing food and cooking) and advanced activities of daily living (establishing and continuing social relationships). Similarly, the psychological evaluation should study the effect of symptoms on self-confidence and self-esteem, and the presence of difficulties in establishing new social relationships (including intimate ones) or loss of previous ones. In addition to applying treatments, the role of professionals should include further interventions, such as: a) providing contrasted and updated information on treatments and interventions with proven evidence-based effects; b) monitoring the sources of information and unproven treatments used by their patients, and early detection of their (harmful) effects, in order to adopt strategies to control their symptoms; c) teaching them to adopt strategies for remembering or controlling the quality of food, and encouraging their involvement in the purchase and preparation of their own meals, to increase their sense of control over their symptoms; d) early detection of "false beliefs" about the unexpected "beneficial" effects of symptoms, such as weight loss. Burges Watson et al. [8] stated that weight loss was commonly reported among Long COVID patients with olfactory/taste disorders, however, this was not always considered to be a problem. Rather, it is thought that it may help certain individuals to begin adopting healthier habits.

General Recommendations

It is critical that the revised manuscript clearly delineate the novel contributions of your research and articulate the unique insights and previously unrecognized knowledge that it brings to the forefront. In addition, the manuscript would benefit from a thorough analysis and synthesis of the collected data, addressing perceived challenges, coping mechanisms, and other important aspects that have not been previously explored.

RESPONSE:

We agree with the reviewer. We have followed your suggestion by making changes (recommended also by other reviewers) to the results and discussion which justify the new findings found in this study. These new findings are included by comparing them specifically with those obtained in other similar studies.

The Strengths and Limitations section has been rewritten as follows:

The strength of this study is that there are few qualitative studies describing the perspective of patients with Long-COVID taste and smell loss and/or disorders [8,27]. The qualitative design enables us to explore and describe the participants’ perspectives in depth and helps us to understand olfactory and taste disorders for people with COVID [8,22-25,27]. Compared to previous studies [8,27] on the perspective of patients with Long COVID with olfactory and taste disorders, our study adds these new findings: a) these symptoms entail losing a gateway to their memories; b) the alteration of taste does not always cause a change in negative dietary habits (more alcohol consumption, less consumption of vegetables), positive changes can also appear (more consumption of fish, less consumption of sweets); c) people who have jobs related to smell and taste do not stop working because of the financial loss involved; d) the possible consequences of the loss of taste are more tangible for the participants, but the symptom that most affects them is the loss of smell; e) they eat with their sight, therefore the presentation of food (and identifying what they eat) becomes more important for these patients and facilitates intake; f) the lack of updated information results in searching alternative sources such as the Internet without being able to verify the information; g) the lack of a clear treatment leads to experimentation with all kinds of treatments, whether or not they are evidence-based; h) professionals should avoid underestimating their symptoms, judging their decisions, or discouraging their attempts at finding a cure during the relationship with the patient.

 La explicación de las diferencias en comparación con los estudios de Almgren et al. y Burges Watson et al. [8,27] podría deberse en parte a que, en el caso de Almgren et al., [27] su estudio se centró en describir la perspectiva de los pacientes sobre la COVID prolongada. Aunque encontraron resultados parciales sobre los síntomas del olfato y el gusto, no se centraron específicamente en ellos. Por el contrario, en el estudio de Burges Watson et al. [8], al describir la perspectiva de los pacientes con COVID prolongado sobre los trastornos del olfato y el gusto, utilizaron el grupo de Facebook COVID-19 Smell and Taste Loss, donde publicaron una serie de preguntas y recibieron publicaciones y respuestas de todos los usuarios. Este sistema podría tener limitaciones a la hora de profundizar en la experiencia individual y/o contrastar otra información como el diagnóstico médico y los síntomas identificados por los profesionales. Además, nuestro estudio se llevó a cabo en un entorno sociosanitario diferente.

Confío en que estas sugerencias le sirvan como guía valiosa para perfeccionar su manuscrito. Agradecemos enormemente su compromiso para mejorar nuestra comprensión de este tema y espero ver la versión mejorada de su artículo.

RESPUESTA:

Estamos muy agradecidos al revisor por los comentarios recibidos. Hemos revisado sus comentarios detenidamente y hemos hecho todo lo posible para abordarlos uno por uno. Esperamos que el manuscrito haya sido mejorado en consecuencia.

Esperamos que esta revisión sea satisfactoria y que el manuscrito sea ahora apto para su publicación en Healthcare.

Atentamente,

Los autores

Reviewer 3 Report

Comments and Suggestions for Authors

Dear authors, I do appreciate your hard work to prepare this manuscript, however I cannot recommend it to the further publication process. There is still lots of things you could do to make this manuscript more scientific. As long as you decided to use qualitative data, you could use Chi-square test (using just Yes/No answers) to perform a proper statistical analysis. Also, I really doubt you could conclude based on the sample of twelve participants....

I do encourage you to perform statistical analysis, redesign this study as a pilot study/preliminary report and once this is done, please resubmit this manuscript.

Comments on the Quality of English Language

Please double check the quality of English

Author Response

Manuscript ID: healthcare-2891731
Type of manuscript: Article
Title: Real-world Experience of Olfactory and Taste Disorders among People with Post-
COVID-19 Condition: A Qualitative Study Using In-Depth Interviews
We would like to thank the Editor and the Reviewers for their careful consideration of our
manuscript. We would also like to thank the Reviewers for their suggestions, which we
believe have enhanced the quality of the manuscript. We have highlighted all the
changes we have made throughout the text in yellow. Below, please find a detailed list
of how we have addressed each comment.
Dear authors, I do appreciate your hard work to prepare this manuscript, however I cannot
recommend it to the further publication process. There is still lots of things you could do
to make this manuscript more scientific.
RESPONSE:
The authors thank the reviewer for these comments. However, we disagree that the article
presented is unscientific or of poor scientific value. The present work belongs to another scientific
paradigm, albeit it is scientific. In medicine different paradigms exist (positivism, post-positivist,
constructivist, critical theory, etc.), and depending on the research question, researchers choose
the most suitable paradigm, together with the methodology (quantitative or qualitative) and the
methods (study procedures) (Varpio & MacLeod A, 2020).
The Academic Medicine Journal published a series of papers on Philosophy of Science between
2020 and 2022,(Varpio & MacLeod A, 2020), showing the different scientific paradigms used for
medical education and research. Among these paradigms, interpretationism is described, as used
in the present study
The aim of this study was to describe the perspective of a group of individuals with long-COVID
taste and smell disorders on their daily life.
As Herbert et al. (2011) pointed out in their work on clinical evidence in health sciences, clinical
evidence is underpinned by asking relevant clinical questions. These questions are oriented
towards effects of intervention, patients’ experiences, the course of a condition (prognosis), and
the accuracy of diagnostic tests (page 10).
Herbert et al. (2011) reported that: “Questions about experiences can relate to any aspect of
clinical practice. Because such questions are potentially very diverse, they must be relatively
open. We recommend that, when formulating questions about experiences, you specify the
patient or problem and the phenomena of interest.” (page 12)
Also, theses authors (Herbert et al., 2011) showed that: “You may also have questions about
which elements of the interventions are the most important and what should be core content. At
the same time you might like to know about the experiences of children attending asthma schools,
and the experiences of the parents of those children, or how you could motivate families from
deprived areas to attend. Most of these questions cannot be answered by clinical trials.
Randomized trials and systematic reviews of randomized trials can tell us whether particular
interventions are effective, but they cannot provide deep insights into patients’ experiences of
those interventions.” (page 26)
In the case of patient experiences, there are various techniques and methodologies that can help
to answer them, such as clinical observation and qualitative studies. In the case of patients'
experiences, qualitative research is recommended.
Herbert et al. (2011) reported that: “Questions about experiences are best answered by qualitative
methods. Qualitative research methods, also called methods of naturalistic inquiry, were
developed in the social and human sciences and refer to theories on interpretation (hermeneutics)
and human experience (phenomenology) (…) are useful for the study of human and social
experience, communication, thoughts, expectations, meanings, attitudes and processes,
especially those related to interaction, relations, development, interpretation, movement and
activity…” (page 27).
Qualitative research is used in a vast number of publications in the field of health sciences,
together with monographic issues explaining its application and use in medical sciences, in
internal medicine (Moser & Korstjens, 2018), rehabilitation (Klem et al., 2021), medical education
(Varpio &, MacLeod, 2020), neurology (Busetto et al., 2020), and neuropaediatrics (Jack &
Phoenix, 2022), among others. By entering “qualitative research” as keyword in pub med/med
line 265,251 results appear.
Reference:
• Busetto L, Wick W, Gumbinger C. How to use and assess qualitative research methods. Neurol
Res Pract. 2020 May 27;2:14. doi: 10.1186/s42466-020-00059-z.
• Herbert R, Jamtvedt G, Birger Hagen K, Mead J. Practical Evidence-Based Physiotherapy. 2 ed.
Elsevier, 2011.
• Jack SM, Phoenix M. Qualitative health research in the fields of developmental medicine and child
neurology. Dev Med Child Neurol. 2022 Jul;64(7):830-839. doi: 10.1111/dmcn.15182.
• Klem NR, Smith A, Shields N, Bunzli S. Demystifying Qualitative Research for Musculoskeletal
Practitioners Part 1: What Is Qualitative Research and How Can It Help Practitioners Deliver Best-
Practice Musculoskeletal Care? J Orthop Sports Phys Ther. 2021 Nov;51(11):531-532. doi:
10.2519/jospt.2021.0110.
• Moser A, Korstjens I. Series: Practical guidance to qualitative research. Part 1: Introduction. Eur J
Gen Pract. 2017 Dec;23(1):271-273. doi: 10.1080/13814788.2017.1375093.
• Varpio L, MacLeod A. Philosophy of Science Series: Harnessing the Multidisciplinary Edge Effect
by Exploring Paradigms, Ontologies, Epistemologies, Axiologies, and Methodologies. Acad Med.
2020 May;95(5):686-689. doi: 10.1097/ACM.0000000000003142.
As long as you decided to use qualitative data, you could use Chi-square test (using just
Yes/No answers) to perform a proper statistical analysis. Also, I really doubt you could
conclude based on the sample of twelve participants....I do encourage you to perform
statistical analysis, redesign this study as a pilot study/preliminary report and once this is
done, please resubmit this manuscript.
RESPONSE:
This work has been carried out using qualitative methodology, within the interpretationist
paradigm. As such, it not a study carried out using quantitative methodology, within the positivist
paradigm, using qualitative variables, which requires the application of a Chi-square test analysis.
Again, with all due respect, a study using qualitative methodology is not the same as the analysis
of qualitative variables typeically performed in a quantitative study.
Qualitative research does not mean that it is a type of research that uses qualitative variables that
are quantifiable and analyzed (Curry et al., 2009; Creswell & Poth, 2018). Essentially, qualitative
research mainly works with qualitative data such as transcribed narrative texts (obtained in the
interviews), images (pictures, photographs), and written documents (letters, diaries). For this
reason we do not understand this observation.
“Qualitative data can be derived from interaction with participants, such as interviews or focus
groups, or as a result of participant observation or analysis of documents or records. Qualitative
data, whether in the form of transcripts or field notes, are generally presented in narrative form.
When derived from interaction with participants the data are presented as verbatim quotations, to
preserve and represent the voice of the participants.” (Carpenter & Suto, 2008, p.32)
Qualitative research is a type of research that is based on studying the experience and
perspective of people in certain situations of health and illness and how they experience the
impact of interventions and health technologies (Curry et al., 2009; Creswell & Poth, 2018;
Carpenter & Suto, 2008; Pope & Mays, 2006). In the PubMed database, in their Medical Subject
Heading (MeSH) section, “qualitative research” is defined as: “Any type of research that employs
nonnumeric information to explore individual or group characteristics, producing findings not
arrived at by statistical procedures or other quantitative means.”
Creswell & Poth (2018) defined qualitative research as: “Qualitative research begins with
assumptions and the use of interpretive/theoretical frameworks that inform the study of research
problems addressing the meaning individuals or groups ascribe to a social or human problem. To
study this problem, qualitative researchers use an emerging qualitative approach to inquiry, the
collection of data in a natural setting sensitive to the people and places under study, and data
analysis that is both inductive and deductive and establishes patterns or themes. The final written
report or presentation includes the voices of participants, the reflexivity of the researcher, a
complex description and interpretation of the problem, and its contribution to the literatura or a
call for change.”(p.8)
It is true that certain qualitative designs (ethnography or qualitative case-studies) (Curry et al.,
2009; Creswell & Poth, 2018; Carpenter & Suto, 2008) or mixed methods exist (Curry & Nuñez-
Smith, 2015), which, besides the use of qualitative data (narrative, visual) from data collection
tools such as in depth interviews, focus groups and observation, also use other tools such as
questionnaires, scales, etc. (Creswell & Poth, 2018; Carpenter & Suto, 2008). However, this is
not the case in this study. Here we used a phenomenological qualitative design, characterized by
gathering the first person perspective and experience of participants, via in-depth interviews
(Curry et al., 2009).
“Qualitative research may be conducted prior to quantitative research to set the direction for
exploration with quantitative methods, or as follow-up to quantitative studies, where it can aid in
interpretation (…) Qualitative research can also help identify which aspects of the intervention are
valued, or not, and why (…) So it can be useful to read both a study evaluating the effects of an
intervention and a complementary study exploring participants’ experiences of the intervention
(…) There are other areas of qualitative research that are highly relevant to practice. Studies that
have as their objective to understand clients’ health-related perceptions and explore patients’
experiences with therapy can be very useful.”(Herbert et al., 2011.p.28-29).
References:
• Carpenter C, Suto M. Qualitative research for occupational and physical therapists: A practical
guide. Oxford: Black-Well Publishing, 2008.
• Creswell JW, Poth CN. Qualitative inquiry and research design. Choosing among five approaches.
4 ed. Thousand Oaks: SAGE, 2018.
• Cohen DJ, Crabtree BF. Evaluative criteria for qualitative research in health care: controversies
and recommendations. Ann Fam Med 2008;6:331–9.
• Curry LA, Nembhard IM, Bradley EH. Qualitative and Mixed Methods Provide Unique Contributions
to Outcomes Research. Circulation. 2009; 119: 1442–1452.
• Curry L, Nuñez-Smith M. Mixed methods in health sciences research. Thousand Oaks, CA: Sage
publications; 2015.
• Herbert R, Jamtvedt G, Hagen KB, Mead J. Practical Evidence-Based Physiotherapy. 2 ed. Elsevier
Churchill Livingstone: Oxford, England, 2011.p. 28-29.
• O'Brien BC, Harris IB, Beckman TJ, Reed DA, Cook DA. Standards for reporting qualitative
research: a synthesis of recommendations. Acad Med. 2014;89(9):1245-1251.
• Pope C, Mays N. Qualitative research in health care. Oxford: Blackwell Publishing; 2006.
• Tong A, Sainsbury P, Craig J. Consolidated criteria for reporting qualitative research (COREQ): a
32-item checklist for interviews and focus groups. Int J Qual Health Care. 2007;19(6):349–57.
• Shenton AK. Strategies for ensuring trustworthiness in qualitative research projects. Education for
Information. 2004;22:63–75.
Comments on the Quality of English Language
Please double check the quality of English
RESPONSE:
Thanks for your comment. The revised manuscript has been sent for revision by a native English
editing service specialized in scientific writing. As a result, edits have been made throughout the
text to improve English Language, grammar and style. Please see proofreading certificate
attached.
We hope that this revision is satisfactory and that the manuscript is now suitable for publication
in Healthcare.
Sincerely,
The Authors

Round 2

Reviewer 2 Report

Comments and Suggestions for Authors

Dear Authors,

I am pleased to inform you that after reviewing the updated manuscript, I have no further specific comments to add. The revisions meet the expectations outlined in my previous comments. The adjustments you have made have significantly improved the clarity and overall impact of the study, and the manuscript has been refined to a standard that effectively communicates the research findings and provides valuable insights to the field. I look forward to the continued impact of your research in the academic community.

Best Regards,

Reviewer 3 Report

Comments and Suggestions for Authors

Dear authors, thank you for all the amendments you have made and the references you included to support your methodology. Although I am still not convinced of this type of science, your response convinced me to recommend this manuscript for publication.